# A novel approach to decision making in rice quality management using interval-valued Pythagorean fuzzy Schweizer and Sklar power aggregation operators

**Ying Wang**[1,2☯]**, Usman Khalid**[3☯]**, Jawad Ali**[4☯]**, Muhammad Ahsan Binyamin** [3☯] *

**1** Software Engineering Institute of Guangzhou, Guangzhou, China, **2** School of Mathematics and Information Science, Guangzhou University, Guangzhou, China, **3** Department of Mathematics, Government College University Faisalabad, Pakistan, **4** Department of Mathematics, Quaid-i-Azam University, Islamabad, Pakistan

☯ These authors contributed equally to this work.
* mahsanbinyamin@gcuf.edu.pk

**Data Availability Statement:** All the relevant data are within the manuscript.

## Abstract

The Pythagorean fuzzy set and interval-valued intuitionistic fuzzy set are the basis of the interval-valued Pythagorean fuzzy set (IVPFS) which offers an effective approach to addressing the complex uncertainty in decision-analysis processes, making it applicable across a broad spectrum of applications. This paper introduces several aggregation operators within the IVPF framework, such as the interval-valued Pythagorean fuzzy SS power weighted average operator, and the interval-valued Pythagorean fuzzy SS power geometric operator using the notion of power aggregation operators through Schweizer and Sklar (SS) operations. The existence of SS t-norms and t-conorms in the IVPF framework for addressing multi-attribute decision-making problems gives the generated operator's ability to make the information aggregation approach more adaptable compared to other current ones. The application of the proposed approach holds the potential to enhance crop yield, optimize resource utilization, and contribute to the overall sustainability of agriculture. Additionally, sensitivity and comparative analyses are provided to clarify the stability and dependability of the results acquired through this approach.

## 1 Introduction

Multi-attribute decision-making (MADM) has emerged as a key field of study in the field of contemporary decision science. In numerous real-world situations, decision-makers could find it difficult to express their choices using exact numerical numbers. In such cases, the introduction of the fuzzy set concept by Zadeh [1] has proven to be a valuable tool for resolving MADM problems [2]. Following this, the intuitionistic fuzzy set (IFS) [3] has emerged, encompassing both belongingness grade (BG) and non-belongingness grade (NG) simultaneously. The literature boasts numerous applications of IFSs [4].

**Funding:** This work was supported by the National Natural Science Foundation of China (No. 62172116) and the Guangzhou Academician and Expert Workstation (No. 2024-D003). The funders played a role in study design, decision to publish, and preparation of the manuscript.

**Competing interests:** The authors have declared that no competing interests exist.

Turksen [5] introduced the concept of the interval-valued fuzzy set (IVFS) in 1986 as an improvement to fuzzy sets. Decision-makers may find it difficult to articulate their thoughts with precise numerical values in real-life circumstances since information is frequently imprecise and insufficient. In 1989, Atanassov [6] presented the IVIFS concept to overcome this issue. Instead of using a precise numerical number, this method represents the membership and non-membership degrees as a range inside [0, 1]. The terms "concentration, dilation, and normalization are the three fundamental operations in an IFS that were first presented by De *et al.* [7] in 2000. Since its introduction, a growing number of scholars have focused on the intuitionistic fuzzy set theory. The mathematician Atanassov investigated certain higher-order fuzzy sets as a result of the technique's subsequent flaws, which were discovered to be numerous.

Several mathematicians have presented multiple varieties of aggregation operators as well as data measurements built on these sets, which are being successfully applied in tackling MADM problems in a variety of contexts. This clearly illustrates the worth of the notions of IFSs and IVIFSs. The another idea of Soft Set (SS) theory was first introduced by Molodtsov [8] as a key instrument for simulating ambiguous and unclear data. To address the ambiguities throughout the study, researchers have employed a variety of conventional equipment, such as the Fuzzy Set (FS) theory, Intuitionistic Fuzzy Set (IFS) theory, etc., extensively [9–13]. All of these approaches, nevertheless, require parameterized tools, making it impossible to use them effectively in real-world situations. Furthermore, one group might give a BG rating of 0.8, while the others assign a NG rating of 0.4. It's clear that when you add these, $0.8 + 0.4 \geq 1$, and when you square them and sum up, $0.8^2 + 0.4^2 \leq 1$. Therefore, this situation cannot be expressed accurately using IFS. Therefore, Pythagorean fuzzy set (PFS) [14], an extension of IFS, has been widely used to handle complicated uncertainty in a variety of decision-making scenarios [15–17]. A difference exists between IFS and PFS, even though both take membership and non-membership degrees into consideration: in PFS, the square sum of these values can be less than or equal to 1, while in IFS, the sum of the membership and non-membership grades can be less than or equal to 1.

Yager [18] expanded on the idea of PFSs by using Pythagorean membership degrees in complex number decision-making. A ranking technique for Pythagorean fuzzy numbers (PFNs) was created by Peng and Yang [19] to handle multicriteria group decision-making issues. Pythagorean fuzzy information's continuity and differentiability were examined by Zhang [20], who also extended PFS to interval-valued Pythagorean fuzzy sets (IVPFSs) [21]. Peng and Yang [22] developed different types of operators, including the interval-valued Pythagorean fuzzy weighted average operator, interval-valued Pythagorean fuzzy weighted geometric operator, interval-valued Pythagorean fuzzy point operators, and interval-valued Pythagorean fuzzy point weighted averaging operators. To deal with decision-making issues, they also suggested the interval-valued Pythagorean fuzzy ELECTRE technique. A maximizing deviation method was developed by Wei and Zhang [21] to solve decision-making difficulties in interval-valued Pythagorean fuzzy scenarios.

The studies on aggregation operators concentrate on two main aspects. Firstly, there is a focus on operational rules. Presently, the majority of aggregation operators relying on IVPFS employ algebraic operational laws, which are a specific instance of Archimedean t-norm and t-conorm (ATT) [23]. The Schweizer and Sklar (SS) t-norms and t-conorms [24] also meet the criteria of Archimedean t-norms and t-conorms. However, SS operations are more flexible than other current methods because of their configurable parameter. Because of their capacity to adjust to individual requirements, users can make decisions that are either optimistic or pessimistic, which helps decision-makers manage risk effectively during the decision-making process. Many researchers have been interested in SS t-norms and t-conorms because of their

versatility. As a result, many aggregation operators based on these operations have been developed in various fuzzy settings.

Using SS t-norms and t-conorms, we extend the IVIF weighted averaging and geometric operators, motivated by the idea of PFSs [25]. A family of efficient aggregation operators—the IVPFSSPA, IVPFSSPWA, IVPFSSPG, and IVPFSSPWG operators—are presented in this study. By merging the power aggregation operator and SS operations, these operators strengthen the information aggregation process in the IVPFS environment.

## 1.1 Motivation

Interval-valued Pythagorean fuzzy sets (IVPFS) expand on the standard Pythagorean fuzzy sets by inserting intervals. This improvement makes it possible to depict uncertainty more accurately and adaptably, especially in situations involving complicated decision-making. Handling uncertainty and inaccuracy in real-life issues requires a deeper comprehension of information, which is provided by the interval values for both BG and NG. Our results will be more trustworthy and dependable since we can more accurately represent the inherent diversity in the expertise and information by utilizing IVPFS.

By utilizing the SS norms and adding a parameter $\eta < 0$, the interval-valued Pythagorean fuzzy Schweizer and Sklar power aggregation operators improve existing aggregation operators. With a more exact and customized arrangement of distinct assessments, this parameter gives the aggregation procedure more fine-tuning capability. When dealing with complicated and varied decision-making difficulties, interval values are especially helpful since they guarantee improved accuracy and handling of uncertainty. The real essence of the actual information can be reflected in more informative and dependable outputs by using these operators.

The rest of this essay is structured as follows:

- Basic ideas about IVPFSs are explained in Section 2. These include the concept's description, its different operations, and the accuracy and scoring functions connected to IVPFSs.

- The various aggregation operators based on SS operations are introduced together with their attributes in Section 3.

- A methodology for solving MADM problems using IVPF SS operators is presented in Section 4.

- An example of rice selection in agriculture is given in Section 5 to show how useful the suggested method is.

- The sensitivity of the introduced operators is evaluated in Section 6 using characteristic weights and the parameter $\eta$.

- We compare our suggested operators with the current alternatives in Section 7.

- Finally, the paper concludes and outlines potential future directions in Section 8.

## 2 Some fundamental ideas

Here, we outline some fundamental ideas on *IVPFSs*.

**Definition 1**. [26] *A fuzzy set (FS) on* $\Psi$ *is offered as follows*:

$$\Im = \{\langle s_i, \mu_\Im(s_i)\rangle | s_i \in \Psi\},$$

*where $\mu_{\mathfrak{I}}(s_i)$ fulfills the constraint*:

$$0 \le \mu_{\mathfrak{I}}(s_i) \le 1$$

**Definition 2**. [18] *A PFS on $\Psi$ is of the form*:

$$\mathfrak{I} = \{\langle s_i, (\mu_{\mathfrak{I}}(s_i), \nu_{\mathfrak{I}}(s_i)) | s_i \in \Psi \rangle\} \tag{1}$$

*granted that $0 \le (\mu_{\mathfrak{I}}(s_i))^2 + (\nu_{\mathfrak{I}}(s_i))^2 \le 1$ wherein $\mu_{\mathfrak{I}}(s_i)$ is referred as BG and $\nu_{\mathfrak{I}}(s_i)$ is referred as NG.*

**Definition 3**. [27] *An IVPFS $\mathfrak{I}$ on a given set $\Psi$ is defined as*

$$\mathfrak{I} = \{\langle s_i, [\mu_{\mathfrak{I}}^{lb}(s_i), \mu_{\mathfrak{I}}^{ub}(s_i)], [\nu_{\mathfrak{I}}^{lb}(s_i), \nu_{\mathfrak{I}}^{ub}(s_i)] \rangle : s_i \in \Psi\}, \tag{2}$$

*where $0 \le \mu_{\mathfrak{I}}^{lb}(s_i) \le \mu_{\mathfrak{I}}^{ub}(s_i) \le 1, 0 \le \nu_{\mathfrak{I}}^{lb}(s_i) \le \nu_{\mathfrak{I}}^{ub}(s_i) \le 1$ and $0 \le (\mu_{\mathfrak{I}}^{ub}(s_i))^2 + (\nu_{\mathfrak{I}}^{ub}(s_i))^2 \le 1$. Here, $\mu_{\mathfrak{I}}(s_i) = [\mu_{\mathfrak{I}}^{lb}(s_i), \mu_{\mathfrak{I}}^{ub}(s_i)]$ and $\nu_{\mathfrak{I}}(s_i) = [\nu_{\mathfrak{I}}^{lb}(s_i), \nu_{\mathfrak{I}}^{ub}(s_i)]$ denote the BG and NG of $s_i \in \Psi$ in form of interval values. In addition, the function $\kappa_{\mathfrak{I}}(s_i) = [\kappa_{\mathfrak{I}}^{lb}(s_i), \kappa_{\mathfrak{I}}^{ub}(s_i)]$ represent the indeterminacy degree of $s_i$ to $\mathfrak{I}$, where $\kappa_{\mathfrak{I}}^{lb}(s_i) = \sqrt{1 - (\mu_{\mathfrak{I}}^{ub}(s_i))^2 - (\nu_{\mathfrak{I}}^{ub}(s_i))^2}$ and $\kappa_{\mathfrak{I}}^{ub}(s_i) = \sqrt{1 - (\mu_{\mathfrak{I}}^{lb}(s_i))^2 - (\nu_{\mathfrak{I}}^{lb}(s_i))^2}$.*

To facilitate computations, an "Interval-Valued Pythagorean Fuzzy Number (*IVPFN*)" is indicated by $\sigma = ([\mu_{\sigma}^{lb}, \mu_{\sigma}^{ub}], [\nu_{\sigma}^{lb}, \nu_{\sigma}^{ub}])$ which satisfies $(\mu_{\sigma}^{ub}(s_i))^2 + (\nu_{\sigma}^{ub}(s_i))^2 \le 1$. There are some special cases of *IVPFS*, given by

**(a)** If $\mu_{\mathfrak{I}}^{lb}(s_i) = \mu_{\mathfrak{I}}^{ub}(s_i)$ and $\nu_{\mathfrak{I}}^{lb}(s_i) = \nu_{\mathfrak{I}}^{ub}(s_i)$ for all $s_i \in \Psi$, then an *IVPFS* reduces to *PFS*.

**(b)** If $\mu_{\mathfrak{I}}^{ub}(s_i) + \nu_{\mathfrak{I}}^{ub}(s_i) \le 1$, then *IVPFS* transforms to *IVIFS*.

**Definition 4**. [27] *Let $\sigma_1 = ([\mu_1^{lb}, \mu_1^{ub}], [\nu_1^{lb}, \nu_1^{ub}])$ and $\sigma_2 = ([\mu_2^{lb}, \mu_2^{ub}], [\nu_2^{lb}, \nu_2^{ub}])$ be two IVPFNs, then the relation between them are described as follows*:

**(i)** $\sigma_1 = \sigma_2$ *iff* $\mu_1^{lb} = \mu_2^{lb}, \mu_1^{ub} = \mu_2^{ub}, \nu_1^{lb} = \nu_2^{lb}$ *and* $\nu_1^{ub} = \nu_2^{ub}$.

**(ii)** $\sigma_1 \prec \sigma_2$ *iff* $\mu_1^{lb} \le \mu_2^{lb}, \mu_1^{ub} \le \mu_2^{ub}, \nu_1^{lb} \ge \nu_2^{lb}$ *and* $\nu_1^{ub} \ge \nu_2^{ub}$.

**Definition 5**. [27] *For any IVPFN $\sigma = ([\mu_{\sigma}^{lb}, \mu_{\sigma}^{ub}], [\nu_{\sigma}^{lb}, \nu_{\sigma}^{ub}])$, the score function $\wp$ of $\sigma$ is provided by*

$$\wp(\sigma) = \frac{1}{2}((\mu_{\sigma}^{lb})^2 + (\mu_{\sigma}^{ub})^2 - (\nu_{\sigma}^{lb})^2 - (\nu_{\sigma}^{ub})^2), \wp(\sigma) \in [-1, 1]. \tag{3}$$

**Definition 6**. [27] *For any IVPFN $\sigma = ([\mu_{\sigma}^{lb}, \mu_{\sigma}^{ub}], [\nu_{\sigma}^{lb}, \nu_{\sigma}^{ub}])$, the accuracy function $\hbar$ of $\sigma$ is provided by*

$$\hbar(\sigma) = \frac{1}{2}((\mu_{\sigma}^{lb})^2 + (\mu_{\sigma}^{ub})^2 + (\nu_{\sigma}^{lb})^2 + (\nu_{\sigma}^{ub})^2), \hbar(\sigma) \in [0, 1]. \tag{4}$$

*To compare any two IVFFNs $\sigma_1$ and $\sigma_2$ that correspond to the accuracy and scoring functions, the following comparing scheme is presented*:

*If $\wp(\sigma_1) > \wp(\sigma_2)$, then $\sigma_1 \succ \sigma_2$.*

*If $\wp(\sigma_1) = \wp(\sigma_2)$, then*

*if $\hbar(\sigma_1) > \hbar(\sigma_2)$, then $\sigma_1 \succ \sigma_2$;*

*if $\hbar(\sigma_1) < \hbar(\sigma_2)$, then $\sigma_1 \prec \sigma_2$;*

*if $\hbar(\sigma_1) = \hbar(\sigma_2)$, then $\sigma_1 = \sigma_2$.*

**Definition 7**. [27] *Suppose* $\sigma = ([\mu^{lb}, \mu^{ub}], [v^{lb}, v^{ub}])$, $\sigma_1 = ([\mu_1^{lb}, \mu_1^{ub}], [v_1^{lb}, v_1^{ub}])$ *and* $\sigma_2 = ([\mu_2^{lb}, \mu_2^{ub}], [v_2^{lb}, v_2^{ub}])$ *be three IVPFNs with* $\gamma > 0$. *Then some basic operations on IVFFNs are presented as follows:*

**(i)** $\sigma_1 \bigcup \sigma_2 = ([max\{\mu_1^{lb}, \mu_2^{lb}\}, max\{\mu_1^{ub}, \mu_2^{ub}\}], [min\{v_1^{lb}, v_2^{lb}\}, min\{v_1^{ub}, v_2^{ub}\}]);$

**(ii)** $\sigma_1 \bigcap \sigma_2 = ([min\{\mu_1^{lb}, \mu_2^{lb}\}, min\{\mu_1^{ub}, \mu_2^{ub}\}], [max\{v_1^{lb}, v_2^{lb}\}, max\{v_1^{ub}, v_2^{ub}\}]);$

**(iii)** $\sigma_1 \oplus \sigma_2 =$
$$([\sqrt{(\mu_1^{lb})^2 + (\mu_2^{lb})^2 - (\mu_1^{lb})^2(\mu_2^{lb})^2}, \sqrt{(\mu_1^{ub})^2 + (\mu_2^{ub})^2 - (\mu_1^{ub})^2(\mu_2^{ub})^2}], [v_1^{lb}v_2^{lb}, v_1^{ub}v_2^{ub}]);$$

**(iv)** $\sigma_1 \otimes \sigma_2 =$
$$([\mu_1^{lb}\mu_2^{lb}, \mu_1^{ub}\mu_2^{ub}], [\sqrt{(v_1^{lb})^2 + (v_2^{lb})^2 - (v_1^{lb})^2(v_2^{lb})^2}, \sqrt{(v_1^{ub})^2 + (v_2^{ub})^2 - (v_1^{ub})^2(v_2^{ub})^2}]);$$

**(v)** $\gamma\sigma = ([\sqrt{1 - (1 - (\mu^{lb})^2)^\gamma}, \sqrt{1 - (1 - (\mu^{ub})^2 S)^\gamma}], [(v^{lb})^\gamma, (v^{ub})^\gamma]);$

**(vi)** $\sigma^\gamma = ([(\mu^{lb})^\gamma, (\mu^{ub})^\gamma], [\sqrt{1 - (1 - (v^{lb})^2)^\gamma}, \sqrt{1 - (1 - (v^{ub})^2)^\gamma}].$

**Remark 1**. *Let us discuss some particular cases of* $\gamma$ *and* $\sigma$ *at* $\gamma\sigma$ *and* $\sigma^\gamma$ *for* $\gamma > 0$:

**(i)** *If* $\sigma = ([1, 1], [0, 0])$, *then by Definition 7, we have*
$\gamma\sigma = ([1, 1], [0, 0])$ *and* $\sigma^\gamma = ([1, 1], [0, 0])$.

**(ii)** *If* $\sigma = ([0, 0], [1, 1])$, *then by Definition 7, we have*
$\gamma\sigma = ([0, 0], [1, 1])$ *and* $\sigma^\gamma = ([0, 0], [1, 1]))$.

**(iii)** *If* $\gamma \to 0$, *then*
$\gamma\sigma \to ([0, 0], [1, 1])$ *and* $\sigma^\gamma \to ([1, 1], [0, 0])$

**(iv)** *If* $\gamma = 1$, *then* $\gamma\sigma = \sigma$ *and* $\sigma^\gamma = \sigma$.

**Theorem 2**. *Suppose* $\sigma = ([\mu^{lb}, \mu^{ub}], [v^{lb}, v^{ub}])$, $\sigma_1 = ([\mu_1^{lb}, \mu_1^{ub}], [v_1^{lb}, v_1^{ub}])$ *and* $\sigma_2 = ([\mu_2^{lb}, \mu_2^{ub}], [v_2^{lb}, v_2^{ub}])$ *be three IVPFNs with* $\gamma, \gamma_1, \gamma_2 > 0$. *Then the following properties hold:*

**(i)** $\sigma_1 \oplus \sigma_2 = \sigma_2 \oplus \sigma_1;$

**(ii)** $\sigma_1 \otimes \sigma_2 = \sigma_2 \otimes \sigma_1;$

**(iii)** $\gamma(\sigma_1 \oplus \sigma_2) = \gamma\sigma_1 \oplus \gamma\sigma_2;$

**(iv)** $\gamma_1\sigma \oplus \gamma_2\sigma = (\gamma_1 \oplus \gamma_2)\sigma;$

**(v)** $(\sigma_1 \otimes \sigma_2)^\gamma = \sigma_1^\gamma \otimes \sigma_2^\gamma;$

**(vi)** $\sigma_1^\gamma \otimes \sigma_2^\gamma = \sigma(\gamma_1 \otimes \gamma_2);$

**(vii)** $\sigma_1^c \oplus \sigma_2^c = (\sigma_1 \otimes \sigma_2)^c;$

**(viii)** $\sigma_1^c \otimes \sigma_2^c = (\sigma_1 \oplus \sigma_2)^c;$

**(ix)** $\sigma_1^c \bigcup \sigma_2^c = (\sigma_1 \bigcap \sigma_2)^c;$

**(x)** $\sigma_1^c \bigcap \sigma_2^c = (\sigma_1 \bigcup \sigma_2)^c;$

**(xi)** $(\sigma^c)^\gamma = (\gamma\sigma)^c;$

**(xii)** $\gamma(\sigma^c) = (\sigma^\gamma)^c;$

**(xiii)** $\sigma_1 \bigcup \sigma_2 = \sigma_2 \bigcup \sigma_1;$

**(xiv)** $\sigma_1 \bigcap \sigma_2 = \sigma_2 \bigcap \sigma_1$;

**(xv)** $\gamma(\sigma_1 \bigcup \sigma_2) = \gamma\sigma_1 \bigcup \gamma\sigma_2$.

*Proof.* It is trivial by [7].

**Definition 8**. [28, 29] *Suppose* $\sigma_1 = ([\mu_1^{lb}, \mu_1^{ub}], [v_1^{lb}, v_1^{ub}])$ *and* $\sigma_2 = ([\mu_2^{lb}, \mu_2^{ub}], [v_2^{lb}, v_2^{ub}])$ *any two IVPFNs. Then,* $d(\sigma_1, \sigma_2)$, *the normalized Hamming distances between* $\sigma_1$ *and* $\sigma_2$, *are defined as*

$$d(\sigma_1, \sigma_2) = \frac{(|(\mu_1^{lb})^2 - (\mu_2^{lb})^2| + |(\mu_1^{ub})^2 - (\mu_2^{ub})^2|) + (|(v_1^{lb})^2 - (v_2^{lb})^2| + |(v_1^{ub})^2 - (v_2^{ub})^2|)}{4} \quad (5)$$

**Definition 9**. [30] *Let* $\sigma_t$, $(t = 1, 2, \ldots, n)$ *represent the collection of aggregated arguments. Then Power Average* (PA) *operator is provided as*

$$PA(\sigma_1, \sigma_2, \ldots, \sigma_n) = \frac{\sum_{t=1}^n (1 + T(\sigma_t))\sigma_t}{\sum_{t=1}^n (1 + T(\sigma_t))}, \quad (6)$$

*where* $T(\sigma_t) = \sum_{t=1, t \neq r}^n \varsigma(\sigma_t, \sigma r)$ *and* $\varsigma(\sigma_t, \sigma r)$ *denoted the support for* $\sigma_t$ *from* $\sigma_r$ *satisfying the conditions:*

- $\varsigma(\sigma_t, \sigma_r) \in [0, 1]$;

- $\varsigma(\sigma_t, \sigma_r) = \varsigma(\sigma_r, \sigma_t)$;

- $\varsigma(\sigma_t, \sigma_r) \geq \varsigma(\sigma_j, \sigma_s)$, *if* $|\sigma_t - \sigma_r| < |\sigma_j - \sigma_s|$, *where* $|.|$ *represent the distance between two IVPFNs.*

**Definition 10**. [30] *Let* $\sigma_t$, $(t = 1, 2, \ldots, n)$ *represent the collection of aggregated arguments. Then Power Geometric* (PG) *operator is provided as*

$$PG(\sigma_1, \sigma_2, \ldots, \sigma_n) = \prod_{t=1}^n \sigma_t^{\frac{1 + T(\sigma_t)}{\sum_{t=1}^n (1 + T(\sigma_t))}}, \quad (7)$$

*where the same meanings as above are conveyed by the notation.*

**Definition 11**. [30, 31] *The SS t-norms and t-conorms are provided by*

- $T(x, y) = (x^\eta + y^\eta - 1)^{1/\eta}$,

- $S(x, y) = 1 - ((1 - x)^\eta + (1 - y)^\eta - 1)^{1/\eta}$, *respectively, where* $x, y \in [0, 1]$ *and* $\eta < 0$.

*Furthermore, SS t-norms and t-conorms modify into algebraic t-norms and t-conorms, respectively, if* $\eta = 0$.

## 3 Development of IVPF SS power aggregation operators

Here, we develop operators for SS t-norms and t-conorms, namely IVPFSSPA, IVPFSSPWA, IVPFSSPG, and IVPFPWG.

**Definition 12**. *Suppose* $\sigma_1 = ([\mu_1^{lb}, \mu_1^{ub}], [v_1^{lb}, v_1^{ub}])$ *and* $\sigma_2 = ([\mu_2^{lb}, \mu_2^{ub}], [v_2^{lb}, v_2^{ub}])$ *be any two IVPFNs with* $\eta < 0$ *and* $\gamma > 0$. *Then, the following are the definitions of SS operating rules based on SS t-norms and t-conorms:*

$$\sigma_1 \oplus_{SS} \sigma_2 = ([\sqrt{S((\mu_1^{lb})^2, (\mu_2^{lb})^2)}, \sqrt{S((\mu_1^{ub})^2, (\mu_2^{ub})^2)}], [\sqrt{T((v_1^{lb})^2, (v_2^{lb})^2)}, \sqrt{T((v_1^{ub})^2, (v_2^{ub})^2)}]) =$$
$$([\sqrt{1 - ((1 - (\mu_1^{lb})^2)^\eta + (1 - (\mu_2^{lb})^2)^\eta - 1)^{1/\eta}},$$

$$\sqrt{1 - ((1 - (\mu_1^{ub})^2)^\eta + (1 - (\mu_2^{ub})^2)^\eta - 1)^{1/\eta}}] \quad,$$

$$[\sqrt{((v_1^{lb})^{2\eta} + (v_2^{lb})^{2\eta} - 1)^{1/\eta}}, \sqrt{((v_1^{ub})^{2\eta} + (v_2^{ub})^{2\eta} - 1)^{1/\eta}}]);$$

$$\gamma\sigma_1 = ([\sqrt{1 - (\gamma(1 - (\mu_1^{lb})^2)^\eta - (\gamma - 1))^{1/\eta}}, \sqrt{1 - (\gamma(1 - (\mu_1^{ub})^2)^\eta - (\gamma - 1))^{1/\eta}}]$$

$$, [\sqrt{(\gamma(v_1^{lb})^{2\eta} - (\gamma - 1))^{1/\eta}}, \sqrt{(\gamma(v_1^{ub})^{2\eta} - (\gamma - 1))^{1/\eta}}]);$$

$$\sigma_1 \otimes_{SS} \sigma_2 = ([\sqrt{T((\mu_1^{lb})^2, (\mu_2^{lb})^2)}, \sqrt{T((\mu_1^{ub})^2, (\mu_2^{ub})^2)}], [\sqrt{S((v_1^{lb})^2, (v_2^{lb})^2)}, \sqrt{S((v_1^{ub})^2, (v_2^{ub})^2)}])$$

$$= ([\sqrt{((\mu_1^{lb})^{2\eta} + (\mu_2^{lb})^{2\eta} - 1)^{1/\eta}}, \sqrt{((\mu_1^{ub})^{2\eta} + (\mu_2^{ub})^{2\eta} - 1)^{1/\eta}}]$$

$$, [\sqrt{1 - ((1 - (v_1^{lb})^2)^\eta + (1 - (v_2^{lb})^2)^\eta - 1)^{1/\eta}}$$

$$, \sqrt{1 - ((1 - (v_1^{ub})^2)^\eta + (1 - (v_2^{ub})^2)^\eta - 1)^{1/\eta}}]);$$

$$\sigma_1^\gamma = ([\sqrt{(\gamma(\mu_1^{lb})^{2\eta} - (\gamma - 1))^{1/\eta}}, \sqrt{(\gamma(\mu_1^{ub})^{2\eta} - (\gamma - 1))^{1/\eta}}]$$

$$, [\sqrt{1 - (\gamma(1 - (v_1^{lb})^2)^\eta - (\gamma - 1))^{1/\eta}}, \sqrt{1 - (\gamma(1 - (v_1^{ub})^2)^\eta - (\gamma - 1))^{1/\eta}}]);$$

**Theorem 3**. *Suppose $\sigma_1 = ([\mu_1^{lb}, \mu_1^{ub}], [v_1^{lb}, v_1^{ub}])$ and $\sigma_2 = ([\mu_2^{lb}, \mu_2^{ub}], [v_2^{lb}, v_2^{ub}])$ be any two IVPFNs with $\eta < 0$ and $n, n_1, n_2 \geq 0$. Then using Definition 10, the following properties hold:*

- $\sigma_1 \oplus_{SS} \sigma_2 = \sigma_2 \oplus_{SS} \sigma_1$;

- $\sigma_1 \otimes_{SS} \sigma_2 = \sigma_2 \otimes_{SS} \sigma_1$;

- $n(\sigma_1 \oplus_{SS} \sigma_2) = n\sigma_1 \oplus_{SS} n\sigma_2$;

- $n_1\sigma_1 \oplus_{SS} n_2\sigma_1 = (n_1 + n_2)\sigma_1$;

- $\sigma_1^{n_1} \otimes \sigma_1^{n_2} = \sigma_1^{n_1 + n_2}$;

- $\sigma_1^{n_1} \otimes \sigma_2^{n_1} = (\sigma_1 \otimes \sigma_2)^{n_1}$;

*Proof.* Theorem 3 is readily proven.

### 3.1 IVPFSSPA aggregation operator

**Definition 13**. [31] *Assume that there are $\sigma_t$, $(t = 1, 2, \ldots, n)$ IVPFNs in this collection. Next, a mapping from $\mathbb{Z}^n \times \mathbb{Z}^n$ to $\mathbb{Z} \times \mathbb{Z}$ is represented by the IVPFSSPA operator, so that*

$$IVPFSSPA(\sigma_1, \sigma_2, \ldots, \sigma_n) = \frac{\overset{n}{\underset{SS\,t=1}{\oplus}} ((1 + T(\sigma_t))\sigma_t)}{\sum_{t=1}^n (1 + T(\sigma_t))}, \tag{8}$$

*where $T(\sigma_t) = \sum_{t=1, t \neq r}^n \varsigma(\sigma_t, \sigma r)$ and $\varsigma(\sigma_t, \sigma r)$ denoted the support for $\sigma_t$ from $\sigma_r$. Based primarily on Definition [13], the following theorem demonstrates that the aggregate value of IVPFNs is likewise a IVPFN.*

**Theorem 4.** *Let us assume that $\sigma_t$, $(t = 1, 2, \ldots, n)$ is a set of IVPFNs such that $\eta < 0$. The aggregated result is still a IVPFN using the IVPFSSPA operator, which fulfills*

$$
\begin{aligned}
IVPFSSPA(\sigma_1, \sigma_2, \ldots, \sigma_n) &= \frac{\overset{n}{\underset{SS\,t=1}{\oplus}} ((1 + T(\sigma_t))\sigma_t)}{\sum_{t=1}^{n}(1 + T(\sigma_t))} \\
&= \left( \left[ \sqrt{1 - (\sum_{t=1}^{n} \xi_t (1 - (\mu_t^{lb})^2)^\eta)^{1/\eta}}, \sqrt{1 - (\sum_{t=1}^{n} \xi_t (1 - (\mu_t^{ub})^2)^\eta)^{1/\eta}} \right] \right. \\
&\quad \left. , \left[ \sqrt{(\sum_{t=1}^{n} \xi_t (v_t^{lb})^{2\eta})^{1/\eta}}, \sqrt{(\sum_{t=1}^{n} \xi_t (v_t^{ub})^{2\eta})^{1/\eta}} \right] \right),
\end{aligned}
\tag{9}
$$

*where $\xi_t = \frac{(1 + T(\sigma_t))}{\sum_{t=1}^{n}(1 + T(\sigma_t))}$, $(t = 1, 2, \ldots, n)$ and $T(\sigma_t) = \sum_{t=1, t \neq r}^{n} \varsigma(\sigma_t, \sigma r)$.*

*Proof.* At first, we need to prove the following result.

For any $\xi = (\xi_1, \xi_2, \ldots, \xi_n)^T$,

$$
\begin{aligned}
&IVPFSSPA(\sigma_1, \sigma_2, \ldots, \sigma_n) \\
&= \left( \left[ \sqrt{1 - (\sum_{t=1}^{n} \xi_t (1 - (\mu_t^{lb})^2)^\eta - \sum_{t=1}^{n} \xi_t + 1)^{1/\eta}}, \right. \right. \\
&\quad \left. \sqrt{1 - (\sum_{t=1}^{n} \xi_t (1 - (\mu_t^{ub})^2)^\eta - \sum_{t=1}^{n} \xi_t + 1)^{1/\eta}} \right] \\
&\quad \left. , \left[ \sqrt{(\sum_{t=1}^{n} \xi_t (v_t^{lb})^{2\eta} - \sum_{t=1}^{n} \xi_t + 1)^{1/\eta}}, \sqrt{(\sum_{t=1}^{n} \xi_t (v_t^{ub})^{2\eta} - \sum_{t=1}^{n} \xi_t + 1)^{1/\eta}} \right] \right)
\end{aligned}
\tag{10}
$$

We use the induction approach to show the given equation.

Take $t = 2$, then

$$
\begin{aligned}
IVPFSSPA(\sigma_1, \sigma_2) &= \frac{\overset{2}{\underset{SS\,t=1}{\oplus}} ((1 + T(\sigma_t))\sigma_t)}{\sum_{t=1}^{2}(1 + T(\sigma_t))} \\
&= \left( \left[ \sqrt{1 - (\sum_{t=1}^{2} \xi_t (1 - (\mu_t^{lb})^2)^\eta - \sum_{t=1}^{2} \xi_t + 1)^{1/\eta}} \right. \right. \\
&\quad \left. , \sqrt{1 - (\sum_{t=1}^{2} \xi_t (1 - (\mu_t^{ub})^2)^\eta - \sum_{t=1}^{2} \xi_t + 1)^{1/\eta}} \right] \\
&\quad , \left[ \sqrt{(\sum_{t=1}^{2} \xi_t (v_t^{lb})^{2\eta} - \sum_{t=1}^{2} \xi_t + 1)^{1/\eta}} \right. \\
&\quad \left. \left. , \sqrt{(\sum_{t=1}^{2} \xi_t (v_t^{ub})^{2\eta} - \sum_{t=1}^{2} \xi_t + 1)^{1/\eta}} \right] \right)
\end{aligned}
$$

Therefore, the result is true for $t = 2$.

Now, suppose the result is true for $t = m$.

$$IVPFSSPA(\sigma_1, \sigma_2, \ldots, \sigma_m) = \left(\left[\sqrt{1 - (\sum_{t=1}^{m}\xi_t(1 - (\mu_t^{lb})^2)^\eta - \sum_{t=1}^{m}\xi_t + 1)^{1/\eta}}\right.\right.$$
$$\left., \sqrt{1 - (\sum_{t=1}^{m}\xi_t(1 - (\mu_t^{ub})^2)^\eta - \sum_{t=1}^{m}\xi_t + 1)^{1/\eta}}\right]$$
$$, \left[\sqrt{(\sum_{t=1}^{m}\xi_t(v_t^{lb})^{2\eta} - \sum_{t=1}^{m}\xi_t + 1)^{1/\eta}}\right.$$
$$\left.\left., \sqrt{(\sum_{t=1}^{m}\xi_t(v_t^{ub})^{2\eta} - \sum_{t=1}^{m}\xi_t + 1)^{1/\eta}}\right]\right)$$

Now, take t = m+1, then by [12],

$$\xi_{m+1}\sigma_{m+1} = \left(\left[\sqrt{1 - (\xi_{m+1}(1 - (\mu_{m+1}^{lb})^2)^\eta - (\xi_{m+1} - 1)^{1/\eta}}\right.\right.$$
$$\left., \sqrt{1 - (\xi_{m+1}(1 - (\mu_{m+1}^{ub})^2)^\eta - (\xi_{m+1} + 1)^{1/\eta}}\right]$$
$$\left., \left[\sqrt{(\xi_{m+1}(v_{m+1}^{lb})^{2\eta} - (\xi_{m+1} - 1)^{1/\eta}}, \sqrt{(\xi_{m+1}(v_{m+1}^{ub})^{2\eta} - (\xi_{m+1} - 1)^{1/\eta}}\right]\right)$$

$$IVPFSSPA(\sigma_1, \sigma_2, \ldots, \sigma_{m+1}) = IVPFSSPA(\sigma_1, \sigma_2, \ldots, \sigma_m) \oplus \xi_{m+1}\sigma_{m+1}$$

$$= \left(\left[\sqrt{1 - ((1 - (1 - (\sum_{t=1}^{m}\xi_t(1 - (\mu_t^{lb})^2)^\eta - \sum_{t=1}^{m}\xi_t + 1)^{1/\eta}))^\eta + (1 - (1 - (\xi_{m+1}(1 - (\mu_{m+1}^{lb})^2)^\eta - (\xi_{m+1} - 1))^{1/\eta})^\eta - 1)^{1/\eta}}\right.\right.$$
$$\left., \sqrt{1 - (1 - (1 - (\sum_{t=1}^{m}\xi_t(1 - (v_t^{lb})^2)^\eta - \sum_{t=1}^{m}\xi_t + 1)^{1/\eta}))^\eta + (1 - (1 - (\xi_{m+1}(1 - (\mu_{m+1}^{ub})^2)^\eta - (\xi_{m+1} - 1)^{1/\eta})^\eta - 1)^{1/\eta}}\right]$$
$$, \left[\sqrt{((\sum_{t=1}^{m}\xi_t(v_t^{lb})^{2\eta} - \sum_{t=1}^{m}\xi_t + 1) + (\xi_{m+1}(v_{m+1}^{lb})^{2\eta} - (\xi_{m+1} - 1)) - 1)^{1/\eta}}\right.$$
$$\left.\left., \sqrt{((\sum_{t=1}^{m}\xi_t(v_t^{ub})^{2\eta} - \sum_{t=1}^{m}\xi_t + 1) + (\xi_{m+1}(v_{m+1}^{ub})^{2\eta} - (\xi_{m+1} - 1)) - 1)^{1/\eta}}\right]\right)$$
$$= \left(\left[\sqrt{1 - (\sum_{t=1}^{m+1}\xi_t(1 - (\mu_t^{lb})^2)^\eta - \sum_{t=1}^{m+1}\xi_t + 1)^{1/\eta}}\right.\right.$$
$$\left., \sqrt{1 - (\sum_{t=1}^{m+1}\xi_t(1 - (\mu_t^{ub})^2)^\eta - \sum_{t=1}^{m+1}\xi_t + 1)^{1/\eta}}\right]$$
$$, \left[\sqrt{(\sum_{t=1}^{m+1}\xi_t(v_t^{lb})^{2\eta} - \sum_{t=1}^{m+1}\xi_t + 1)^{1/\eta}}, \sqrt{(\sum_{t=1}^{m+1}\xi_t(v_t^{ub})^{2\eta} - \sum_{t=1}^{m+1}\xi_t + 1)^{1/\eta}}\right]\right)$$

Thus, the result is true for $t = m + 1$.

Given that the statement holds true for any value of $\xi$, it will also hold when the following criteria are met: $\xi_t \geq 0$ with $\sum_{t=1}^{n} \xi_t = 1$.

Thus, it is demonstrated that the theorem.

**Theorem 5**. (*Idempotency*) *If all* $\sigma_t = ([\mu_t^{lb}, \mu_t^{ub}], [v_t^{lb}, v_t^{ub}])$ *are equal and* $\sigma_t = \sigma = ([\mu^{lb}, \mu^{ub}], [v^{lb}, v^{ub}])$ *for all* $t = 1, 2, \ldots, n$, *then*

$$IVPFSSPA(\sigma_1, \sigma_2, \ldots, \sigma_n) = \sigma.$$

*Proof.* $IVPFSSPA(\sigma_1, \sigma_2, \ldots, \sigma_n) = \left( \left[ \sqrt{1 - (\sum_{t=1}^n \xi_t(1 - (\mu_t^{lb})^2)^\eta)^{1/\eta}}, \right. \right.$

$\left. \sqrt{1 - (\sum_{t=1}^n \xi_t(1 - (\mu_t^{ub})^2)^\eta)^{1/\eta}} \right], \left[ \sqrt{(\sum_{t=1}^n \xi_t(v_t^{lb})^{2\eta})^{1/\eta}}, \sqrt{(\sum_{t=1}^n \xi_t(v_t^{ub})^{2\eta})^{1/\eta}} \right] \right) =$

$\left( \left[ \sqrt{1 - (\sum_{t=1}^n \xi_t(1 - (\mu^{lb})^2)^\eta)^{1/\eta}}, \sqrt{1 - (\sum_{t=1}^n \xi_t(1 - (\mu^{ub})^2)^\eta)^{1/\eta}} \right] \right.$

$\left. , \left[ \sqrt{(\sum_{t=1}^n \xi_t(v^{lb})^{2\eta})^{1/\eta}}, \sqrt{(\sum_{t=1}^n \xi_t(v^{ub})^{2\eta})^{1/\eta}} \right] \right) = ([\mu^{lb}, \mu^{ub}], [v^{lb}, v^{ub}]) = \sigma.$

**Theorem 6**. (*Monotonicity*) *Consider* $\sigma_t = ([\mu_t^{lb}, \mu_t^{ub}], [v_t^{lb}, v_t^{ub}])$ *and*
$\sigma_t' = ([(\mu^{lb})_t', (\mu^{ub})_t'], [(v^{lb})_t', (v^{ub})_t']), (t = 1, 2, \ldots, n)$ *as two collections of IVPFNs such that*
$\mu_t^{lb} \le (\mu^{lb})_t', \mu_t^{ub} \le (\mu^{ub})_t'; v_t^{lb} \ge (v^{lb})_t', v_t^{ub} \ge (v^{ub})_t'$ *for all* $t \in 1, 2, \ldots, n$, *then*

$$IVPFSSPA(\sigma_1, \sigma_2, \ldots, \sigma_t) \le IVPFSSPA(\sigma_1', \sigma_2', \ldots, \sigma_t').$$

*Proof.* Since $\mu_t^{lb} \le (\mu^{lb})_t'$ and $v_t^{lb} \ge (v^{lb})_t'$, $\Rightarrow (1 - (\mu_t^{lb})^2)^\eta \le (1 - ((\mu^{lb})_t')^2)^\eta$, (Since $\eta < 0$)
$\Rightarrow (\sum_{t=1}^n \xi_t'(1 - ((\mu^{lb})_t')^2)^\eta))^{1/\eta} \le (\sum_{t=1}^n \xi_t(1 - (\mu_t^{lb})^2)^\eta))^{1/\eta}$,
$\Rightarrow 1 - (\sum_{t=1}^n \xi_t(1 - (\mu_t^{lb})^2)^\eta))^{1/\eta} \le 1 - (\sum_{t=1}^n \xi_t'(1 - ((\mu^{lb})_t')^2)^\eta))^{1/\eta}$ and
$(\sum_{t=1}^n \xi_t'((v^{lb})_t'))^{2\eta} \le (\sum_{t=1}^n \xi_t(v_t^{lb}))^{2\eta}$.

Similarly, we have $\Rightarrow 1 - (\sum_{t=1}^n \xi_t(1 - (\mu_t^{ub})^2)^\eta))^{1/\eta} \le 1 - (\sum_{t=1}^n \xi_t'(1 - ((\mu^{ub})_t')^2)^\eta))^{1/\eta}$
and $(\sum_{t=1}^n \xi_t'((v^{ub})_t'))^{2\eta} \le (\sum_{t=1}^n \xi_t(v_t^{ub}))^{2\eta}$.

Therefore, $IVPFSSPA(\sigma_1, \sigma_2, \ldots, \sigma_t) \le IVPFSSPA(\sigma_1', \sigma_2', \ldots, \sigma_t')$.

**Theorem 7**. (*Boundedness*) *Let* $\sigma_t = ([\mu_t^{lb}, \mu_t^{ub}], [v_t^{lb}, v_t^{ub}])$ $(t = 1, 2, \ldots, n)$ *be an array of IVPFNs, then*

$$\sigma_{min} \le IVPFSSPA(\sigma_1, \sigma_2, \ldots, \sigma_t) \le \sigma_{max}$$

*where* $\sigma_{min} = \min_t \{\sigma_t\}$ *and* $\sigma_{max} = \max_t \{\sigma_t\}$.

*Proof.* $\mu_{min}^{lb} = \min_t(\mu_t^{lb}), \mu_{min}^{ub} = \min_t(\mu_t^{ub}), v_{min}^{lb} = \min_t(v_t^{lb}), v_{min}^{ub} = \min_t(v_t^{ub})$;
$\mu_{max}^{lb} = \max_t(\mu_t^{lb}), \mu_{max}^{ub} = \max_t(\mu_t^{ub}), v_{max}^{lb} = \max_t(v_t^{lb}), v_{max}^{ub} = \max_t(v_t^{ub})$.
Assume that $IVPFSSPA(\sigma_1, \sigma_2, \ldots, \sigma_n) = \sigma = ([\mu^{lb}, \mu^{ub}], [v^{lb}, v^{ub}])$.
Then obviously

$$([\mu_{min}^{lb}, \mu_{min}^{ub}], [v_{max}^{lb}, v_{max}^{ub}]) \le ([\mu^{lb}, \mu^{ub}], [v^{lb}, v^{ub}]) \tag{11}$$

$$([\mu_{max}^{lb}, \mu_{max}^{ub}], [v_{min}^{lb}, v_{min}^{ub}]) \ge ([\mu^{lb}, \mu^{ub}], [v^{lb}, v^{ub}]). \tag{12}$$

Thus, from (11) and (12), we have $\sigma_{min} \le IVPFSSPA(\sigma_1, \sigma_2, \ldots, \sigma_t) \le \sigma_{max}$.

**Remark 8**. *When* $\eta = 0$, *then particular form of IVPFSSPA is originate, that is, IVPFPA operator.*

*In this case,*

$$
\begin{aligned}
IVPFPA_{\eta=0}(\sigma_1, \sigma_2, \ldots, \sigma_n) \quad &= \left( \left[ \sqrt{1 - \prod_{t=1}^{n}(1-(\mu_t^{lb})^2)^{\xi_t}}, \sqrt{1 - \prod_{t=1}^{n}(1-(\mu_t^{ub})^2)^{\xi_t}} \right] \right. \\
&\left. , \left[ \prod_{t=1}^{n}(v_t^{lb})^{\xi_t}, \prod_{t=1}^{n}(v_t^{ub})^{\xi_t} \right] \right),
\end{aligned}
$$

*where* $\xi_t = \frac{(1+T(\sigma_t))}{\sum_{t=1}^{n}(1+T(\sigma_t))}, (t = 1, 2, \ldots, n).$

## 3.2 IVPFSSPWA aggregation operator

**Definition 14.** [31] *Suppose* $\sigma_t$, $(t = 1, 2, \ldots, n)$ *be a collection of IVPFNs and* $\tilde{\omega} = (\omega_1, \omega_2, \ldots, \omega_n)^T$ *be the weight vector in which* $\omega_t \in [0, 1]$ *for* $t \in 1, 2, \ldots, n$ *and* $\sum_{t=1}^{n} \omega_t = 1$. *Then, a mapping from* $\mathbb{Z}^n \times \mathbb{Z}^n$ *to* $\mathbb{Z} \times \mathbb{Z}$ *is represented by the IVPFSSPWA operator, so that*

$$
IVPFSSPWA(\sigma_1, \sigma_2, \ldots, \sigma_n) = \frac{\overset{n}{\underset{SS \, t=1}{\oplus}} (\omega_t(1 + T(\sigma_t))\sigma_t)}{\sum_{t=1}^{n} \omega_t(1 + T(\sigma_t))}, \tag{13}
$$

*where* $T(\sigma_t) = \sum_{t=1, t \neq r}^{n} \varsigma(\sigma_t, \sigma r)$ *and* $\varsigma(\sigma_t, \sigma r)$ *denoted the support for* $\sigma_t$ *from* $\sigma_r$.

**Theorem 9.** *Suppose* $\sigma_t$, $(t = 1, 2, \ldots, n)$ *be a collection of IPFFNs with* $\eta < 0$. *Let* $\tilde{\omega} = (\omega_1, \omega_2, \ldots, \omega_n)^T$ *be the weight vector in which* $\omega_t \in [0, 1]$ *for* $t \in 1, 2, \ldots, n$ *and* $\sum_{t=1}^{n} \omega_t = 1$. *Then, using IVPFSSPA operator, the aggregated result is still a IVPFN which satisfies*

$$
\begin{aligned}
IVPFSSPWA(\sigma_1, \sigma_2, \ldots, \sigma_n) &= \frac{\overset{n}{\underset{SS \, t=1}{\oplus}} (\omega_t(1 + T(\sigma_t))\sigma_t)}{\sum_{t=1}^{n} \omega_t(1 + T(\sigma_t))} \\
&= \left( \left[ \sqrt{1 - (\sum_{t=1}^{n}\lambda_t(1-(\mu_t^{lb})^2)^{\eta})^{1/\eta}}, \sqrt{1 - (\sum_{t=1}^{n}\lambda_t(1-(\mu_t^{ub})^2)^{\eta})^{1/\eta}} \right] \right. \\
&\left. , \left[ \sqrt{(\sum_{t=1}^{n}\lambda_t(v_t^{lb})^{2\eta})^{1/\eta}}, \sqrt{(\sum_{t=1}^{n}\lambda_t(v_t^{ub})^{2\eta})^{1/\eta}} \right] \right)
\end{aligned} \tag{14}
$$

*where* $\lambda_t = \frac{\omega_t(1+T(\sigma_t))}{\sum_{t=1}^{n}\omega_t(1+T(\sigma_t))}, (t = 1, 2, \ldots, n)$ *and* $T(\sigma_t) = \sum_{t=1, t \neq r}^{n} \varsigma(\sigma_t, \sigma r)$.

*Proof.* This proof is the same as the proof of Theorem 4.

**Remark 10.** *The properties of the IVPFSSPWA operator are likewise satisfied (idempotency, monotonicity, and boundedness), and they can be proved similarly to the IVPFSSPA operator scenarios.*

**Remark 11.** *When* $\eta = 0$, *then particular form of IVPFSSPWA is originate, that is, IVPFPWA operator.*

*In this case,*

$$
\begin{aligned}
IVPFPWA_{\eta=0}(\sigma_1, \sigma_2, \ldots, \sigma_n) \quad &= \left( \left[ \sqrt{1 - \prod_{t=1}^{n}(1-(\mu_t^{lb})^2)^{\lambda_t}}, \sqrt{1 - \prod_{t=1}^{n}(1-(\mu_t^{ub})^2)^{\lambda_t}} \right] \right. \\
&\left. , \left[ \prod_{t=1}^{n}(v_t^{lb})^{\lambda_t}, \prod_{t=1}^{n}(v_t^{ub})^{\lambda_t} \right] \right),
\end{aligned}
$$

*where* $\lambda_t = \frac{\omega_t(1+T(\sigma_t))}{\sum_{t=1}^{n}\omega_t(1+T(\sigma_t))}, (t = 1, 2, \ldots, n).$

### 3.3 IVPFSSPG aggregation operator

**Definition 15.** [31] *Assume that there are $\sigma_t$, ($t = 1, 2, \ldots, n$) IVPFNs in this collection. Next, a mapping from $\mathbb{Z}^n \times \mathbb{Z}^n$ to $\mathbb{Z} \times \mathbb{Z}$ is represented by the IVPFSSPG operator, so that*

$$IVPFSSPG(\sigma_1, \sigma_2, \ldots, \sigma_n) = \overset{n}{\underset{t=1}{\otimes}}((\sigma_t)^{\frac{1+T(\sigma_t)}{\sum_{t=1}^{n}(1+T(\sigma_t))}}), \tag{15}$$

*where $T(\sigma_t) = \sum_{t=1, t \neq r}^{n} \varsigma(\sigma_t, \sigma r)$ and $\varsigma(\sigma_t, \sigma r)$ denoted the support for $\sigma_t$ from $\sigma_r$. The next theorem is proposed by using Definition [15].*

**Theorem 12.** *Suppose $\sigma_t$, ($t = 1, 2, \ldots, n$) be a collection of IVPFNs with $\eta < 0$. Then, using IVPFSSPG operator, the aggregated result is still a IVPFN which satisfies*

$$
\begin{aligned}
IVPFSSPG(\sigma_1, \sigma_2, \ldots, \sigma_n) \quad &= \overset{n}{\underset{t=1}{\otimes}}((\sigma_t)^{\frac{1+T(\sigma_t)}{\sum_{t=1}^{n}(1+T(\sigma_t))}}) \\
&= \left( \left[ \sqrt{(\sum_{t=1}^{n} \xi_t (\mu_t^{lb})^{2\eta})^{1/\eta}}, \sqrt{(\sum_{t=1}^{n} \xi_t (\mu_t^{ub})^{2\eta})^{1/\eta}} \right] \right. \\
&\quad \left. , \left[ \sqrt{1 - (\sum_{t=1}^{n} \xi_t (1 - (v_t^{lb})^2)^\eta)^{1/\eta}} \right. \right. \\
&\quad \left. \left. , \sqrt{1 - (\sum_{t=1}^{n} \xi_t (1 - (v_t^{ub})^2)^\eta)^{1/\eta}} \right] \right)
\end{aligned} \tag{16}
$$

*where $\xi_t = \frac{(1+T(\sigma_t))}{\sum_{t=1}^{n}(1+T(\sigma_t))}, (t = 1, 2, \ldots, n)$ and $T(\sigma_t) = \sum_{t=1, t \neq r}^{n} \varsigma(\sigma_t, \sigma r)$.*

*The proof of this theorem is on the lines of the above proved theorem.*

**Theorem 13.** *(Idempotency) If all $\sigma_t = ([\mu_t^{lb}, \mu_t^{ub}], [v_t^{lb}, v_t^{ub}])$ are equal and $\sigma_t = \sigma = ([\mu^{lb}, \mu^{ub}], [v^{lb}, v^{ub}])$ for all $t = 1, 2, \ldots, n$, then*

$$IVPFSSPG(\sigma_1, \sigma_2, \ldots, \sigma_n) = \sigma.$$

**Theorem 14.** *(Monotonicity) Consider $\sigma_t = ([\mu_t^{lb}, \mu_t^{ub}], [v_t^{lb}, v_t^{ub}])$ and $\sigma'_t = ([(\mu^{lb})'_t, (\mu^{ub})'_t], [(v^{lb})'_t, (v^{ub})'_t]), (t = 1, 2, \ldots, n)$ as two collections of IVPFNs such that $\mu_t^{lb} \leq (\mu^{lb})'_t, \mu_t^{ub} \leq (\mu^{ub})'_t; v_t^{lb} \geq (v^{lb})'_t, v_t^{ub} \geq (v^{ub})'_t$ for all $t \in 1, 2, \ldots, n$, then*

$$IVPFSSPG(\sigma_1, \sigma_2, \ldots, \sigma_t) \leq IVPFSSPG(\sigma'_1, \sigma'_2, \ldots, \sigma'_t).$$

**Theorem 15.** *(Boundedness) Let $\sigma_t = ([\mu_t^{lb}, \mu_t^{ub}], [v_t^{lb}, v_t^{ub}])$ ($t = 1, 2, \ldots, n$) be an array of IVPFNs, then*

$$\sigma_{min} \leq IVPFSSPG(\sigma_1, \sigma_2, \ldots, \sigma_t) \leq \sigma_{max}.$$

**Remark 16.** *When $\eta = 0$, then particular form of IVPFSSPG is originate, that is, IVPFPG operator.*

*In this case,*

$$
IVPFPG_{\eta=0}(\sigma_1, \sigma_2, \ldots, \sigma_n) = \left( \left[ \prod_{t=1}^{n}(\mu_t^{lb})^{\xi_t}, \prod_{t=1}^{n}(\mu_t^{ub})^{\xi_t} \right] \right.
$$
$$
\left. , \left[ \sqrt{1 - \prod_{t=1}^{n}(1-(v_t^{lb})^2)^{\xi_t}}, \sqrt{1 - \prod_{t=1}^{n}(1-(v_t^{ub})^2)^{\xi_t}} \right] \right),
$$

*where* $\xi_t = \frac{(1+T(\sigma_t)}{\sum_{t=1}^{n}(1+T(\sigma))}, (t=1,2,\ldots,n).$

## 3.4 IVPFSSPWG aggregation operator

**Definition 16**. [31] *Suppose* $\sigma_t$, $(t = 1, 2, \ldots, n)$ *be a collection of IVPFNs and* $\tilde{\omega} = (\omega_1, \omega_2, \ldots, \omega_n)^T$ *be the weight vector in which* $\omega_t \in [0, 1]$ *for* $t \in 1, 2, \ldots, n$ *and* $\sum_{t=1}^{n} \omega_t = 1$. *Then, a mapping from* $\mathbb{Z}^n \times \mathbb{Z}^n$ *to* $\mathbb{Z} \times \mathbb{Z}$ *is represented by the IVPFSSPWG operator, so that*

$$
IVPFSSPWG(\sigma_1, \sigma_2, \ldots, \sigma_n) = \overset{n}{\underset{t=1}{\otimes}} ((\sigma_t)^{\frac{\omega_t(1+T(\sigma_t))}{\sum_{t=1}^{n}\omega_t(1+T(\sigma_t))}}), \tag{17}
$$

*where* $T(\sigma_t) = \sum_{t=1, t \neq r}^{n} \varsigma(\sigma_t, \sigma r)$ *and* $\varsigma(\sigma_t, \sigma r)$ *denoted the support for* $\sigma_t$ *from* $\sigma_r$.

 **Theorem 17**. *Suppose* $\sigma_t$, $(t = 1, 2, \ldots, n)$ *be a collection of IVPFNs with* $\eta < 0$. *Let* $\tilde{\omega} = (\omega_1, \omega_2, \ldots, \omega_n)^T$ *be the weight vector in which* $\omega_t \in [0, 1]$ *for* $t = 1, 2, \ldots, n$ *and* $\sum_{t=1}^{n} \omega_t = 1$. *Then, using IVPFSSPWG operator, the aggregated result is still a IVPFN which satisfies*

$$
IVPFSSPWG(\sigma_1, \sigma_2, \ldots, \sigma_n) = \overset{n}{\underset{t=1}{\otimes}} ((\sigma_t)^{\frac{\omega_t(1+T(\sigma_t))}{\sum_{t=1}^{n}\omega_t(1+T(\sigma_t))}})
$$
$$
\left( \left[ \sqrt{(\sum_{t=1}^{n}\lambda_t(\mu_t^{lb})^{2\eta})^{1/\eta}}, \sqrt{(\sum_{t=1}^{n}\lambda_t(\mu_t^{ub})^{2\eta})^{1/\eta}} \right] \right.
$$
$$
, \left[ \sqrt{1 - (\sum_{t=1}^{n}\lambda_t(1-(v_t^{lb})^2)^{\eta})^{1/\eta}} \right.
$$
$$
\left. \left. , \sqrt{1 - (\sum_{t=1}^{n}\lambda_t(1-(v_t^{ub})^2)^{\eta})^{1/\eta}} \right] \right) \tag{18}
$$

*where* $\lambda_t = \frac{\omega_t(1+T(\sigma_t)}{\sum_{t=1}^{n}\omega_t(1+T(\sigma_t))}, (t=1,2,\ldots,n)$ *and* $T(\sigma_t) = \sum_{t=1, t\neq r}^{n} \varsigma(\sigma_t, \sigma r)$.

 **Remark 18**. *The IVPFSSPWG operator also satisfies properties like Idempotency, Monotonicity and Boundedness.*

 **Remark 19**. *When* $\eta = 0$, *then particular form of IVPFSSPWG is originate, that is, IVPFPWG operator.*

 *In this case,*

$$
IVPFPWG_{\eta=0}(\sigma_1, \sigma_2, \ldots, \sigma_n) = \left( \left[ \prod_{t=1}^{n}(\mu_t^{lb})^{\lambda_t}, \prod_{t=1}^{n}(\mu_t^{ub})^{\lambda_t} \right] \right.
$$
$$
\left. , \left[ \sqrt{1 - \prod_{t=1}^{n}(1-(v_t^{lb})^2)^{\lambda_t}}, \sqrt{1 - \prod_{t=1}^{n}(1-(v_t^{ub})^2)^{\lambda_t}} \right] \right),
$$

*where* $\lambda_t = \frac{\omega_t(1+T(\sigma_t)}{\sum_{t=1}^{n}\omega_t(1+T(\sigma_t))}, (t=1,2,\ldots,n).$

## 4 Method for using IVPF SS operators to solve MADM problems

The goal of this division is to concentrate the *MADM* approach around the recently established *IVPFSS* power aggregation operators. The set of alternatives for each choice is denoted by R = {$R_1$, $R_2$, ..., $R_m$}, and the set of attributes with the weight vector $\tilde{\omega} = \{\omega_1, \omega_2, ..., \omega_n\}$ is denoted by J = {$J_1$, $J_2$, ..., $J_n$}. For $\sum_{t=1}^{n} \tilde{\omega}_t = 1$ and $\tilde{\omega}_k \in [0, 1]$, the relative weights of various criteria in the decision-making process are indicated by the weight vector $\tilde{\omega}$.

Suppose that $\tilde{U} = (\tilde{u_{kt}})_{m \times n} = ([\mu_{kt}^{lb}, \mu_{kt}^{ub}], [v_{kt}^{lb}, v_{kt}^{ub}])$ is the *IVFF* decision matrix in which $\mu_{kt}$ represents the degree to which the alternatives $R_k$ satisfy the $J_t$ attribute, while $v_{kt}$ indicates the degree to which the alternatives $R_k$ do not satisfy the $J_t$ qualities such that $\mu_{kt}$, $v_{kt} \in [0, 1]$ and $0 \le (\mu_{kt}^{ub})^2 + (v_{kt}^{ub})^2 \le 1$, given by decision maker. To find the optimal solutions, the *IVPFSSPWA* and *IVPFSSPWG* operators are now used for the previously mentioned *MADM* issues (see in Fig 1). The following steps are mentioned to solve *MADM* problems:

**Step 1.** *IVPF* decision matrix creation: Given the previously given situation, the *MADM* problem can be expressed in the decision matrix that is subsequently created:

$$\tilde{U}_{m \times n} = \begin{pmatrix} \tilde{u}_{11} & \dots & \tilde{u}_{1g} & \dots & \tilde{u}_{1n} \\ \vdots & \ddots & \vdots & \ddots & \vdots \\ \tilde{u}_{i1} & \dots & \tilde{u}_{ig} & \dots & \tilde{u}_{in} \\ \vdots & \ddots & \vdots & \ddots & \vdots \\ \tilde{u}_{m1} & \dots & \tilde{u}_{mg} & \dots & \tilde{u}_{mn} \end{pmatrix} \quad (19)$$

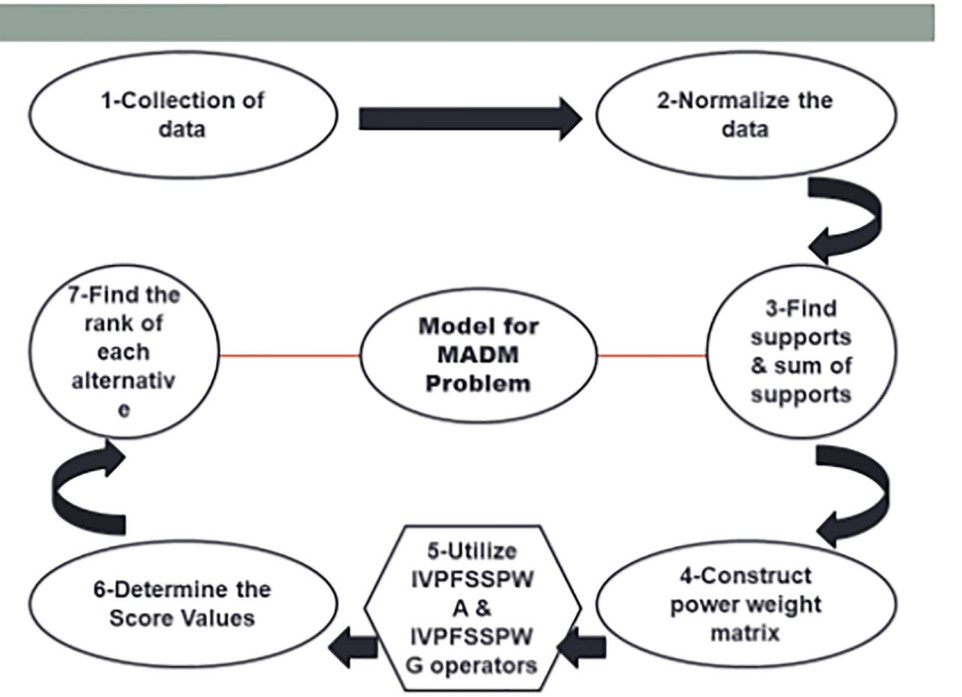

**Fig 1. Graphical models for solving MADM problems.**

**Step 2.** Normality: At this stage, a few benefit and expense criteria are used to modify the *IVPF* assessment matrix $\tilde{U}_{m \times n}$. The benefit criterion performs best and the cost criterion performs poorly when the value grows because these two criteria have opposite responses. Consequently, we apply the next normality approach to ensure that all requirements are met and translate the cost criterion into the benefit criteria.

**Step 3.** Calculate supports of $\sigma_{kt}$ from $\sigma_{kl}(t \neq l)$ for each $k \in 1, 2, \ldots, m$ to determine the proximity of the attributes values of the normalized decision matrix $\tilde{U}$, using Eq (9) as

$$\varsigma(\tilde{\sigma}_{kt}, \tilde{\sigma}_{kl}) = 1 - d(\tilde{\sigma}_{kt}, \tilde{\sigma}_{kl})$$

$$= 1 - \frac{\left(|(\mu_{kt}^{lb})^2 - (\mu_{kl}^{lb})^2| + |(\mu_{kt}^{ub})^2 - (\mu_{kl}^{ub})^2|\right) + \left(|(v_{kt}^{lb})^2 - (v_{kl}^{lb})^2| + |(v_{kt}^{ub})^2 - (v_{kl}^{ub})^2|\right)}{4}$$

Therefore, build the matrix of support

$$\mathbb{S} = [(\mathbb{S}_{tl}^k : t \neq l, l = 1, 2, \ldots, n)],$$

where all supports for $\sigma_{kt}$ from $\sigma_{kl}$ is denoted as a vector of (n-1) tuples having the form $\mathbb{S}_{tl}^k = (\varsigma(\sigma_{kt}, \sigma_{kl})_{l=1, l \neq t}^n$.
Now, find the sum of supports

$$T(\tilde{\sigma}_{kt}) = \sum_{l=1, l \neq t}^{n} \mathbb{S}_{tl}^k,$$

for $k = 1, 2, \ldots, m$ and $t = 1, 2, \ldots, n$.

**Step 4.** Find the weighted closeness of each alternative concerning criteria, by using the known weights $\omega_t(t = 1, 2, \ldots, n)$ corresponding to the attribute $S_t$ as

$$\lambda_{kt} = \frac{\omega_t(1 + T(\sigma_{kt})}{\sum_{t=1}^n \omega_t(1 + T(\sigma_{kt}))}, (t = 1, 2, \ldots, n)$$

Thus, the power weight matrix is produced.

**Step 5.** Aggregation: Aggregate all attribute values corresponding to each alternative by using the *IVPFSSPWA* operator (or, *IVPFSSPWG* operator) for any $\eta < 0$ and get a single *IVPFN* for each alternative by

$$\tilde{\sigma}_k = IVPFSSPWA(\tilde{\sigma}_{k1}, \tilde{\sigma}_{k2}, \ldots, \tilde{\sigma}_{kn}) = \frac{\overset{n}{\underset{SS_{t=1}}{\oplus}} \left(\omega_t(1 + T(\tilde{\sigma}_{kt}))\tilde{\sigma}_{kt}\right)}{\sum_{t=1}^n \omega_t(1 + T(\tilde{\sigma}_{kt}))}$$

$$= \left(\left[\sqrt{1 - (\sum_{t=1}^n \lambda_t(1 - (\mu_t^{lb})^2)^\eta)^{1/\eta}}, \sqrt{1 - (\sum_{t=1}^n \lambda_{kt}(1 - (\mu_{kt}^{ub})^2)^\eta)^{1/\eta}}\right]\right. \tag{20}$$

$$\left., \left[\sqrt{(\sum_{t=1}^n \lambda_{kt}(v_{kt}^{lb})^{2\eta})^{1/\eta}}, \sqrt{(\sum_{t=1}^n \lambda_{kt}(v_{kt}^{ub})^{2\eta})^{1/\eta}}\right]\right)$$

Or,

$$\sigma_k = IVPFSSPWG(\sigma_{k1}, \sigma_{k2}, \ldots, \sigma_{kn}) = \overset{n}{\underset{t=1}{\otimes}}((\sigma_t)^{\frac{\omega_t(1+T(\sigma_{kt}))}{\sum_{t=1}^{n}\omega_t(1+T(\sigma_{kt}))}})$$

$$\left( \left[ \sqrt{(\sum_{t=1}^{n}\lambda_{kt}(\mu_{kt}^{lb})^{2\eta})^{1/\eta}}, \sqrt{(\sum_{t=1}^{n}\lambda_{kt}(\mu_{kt}^{ub})^{2\eta})^{1/\eta}} \right] \right. \tag{21}$$

$$\left. , \left[ \sqrt{1 - (\sum_{t=1}^{n}\lambda_{kt}(1 - (v_{kt}^{lb})^2)^{\eta})^{1/\eta}}, \sqrt{1 - (\sum_{t=1}^{n}\lambda_{kt}(1 - (v_{kt}^{ub})^2)^{\eta})^{1/\eta}} \right] \right)$$

**Step 6.** Score values: From (3), the score value of the concluding $IVPFNs$ $\tilde{\sigma}_k(k = 1, 2, \ldots, m)$ are find. When score values are the same then we use the accuracy function instead of the score function from 4.

**Step 7.** In this last phase, each alternative is ranked according to score values (when same then accuracy function), and the best-matched answer is selected.

## 5 Exemplary case study

In this section, the example of deciding the growth of rice is used to elaborate on the implications as well as the practicality of the suggested method. It is essential that the suggested method can be applied to a range of dynamic problems and is not just restricted to the rice determination problem. By considering multiple attributes and providing a structured approach to decision-making, these systems empower farmers, researchers, and policymakers to make choices that improve crop yield, resource utilization, and overall sustainability of agriculture. They make calls to those who are skilled in determining rice. The observes that some standards are used to examine the growth of rice: Rainfall($J_1$), Soil Quality($J_2$), Temperature ($J_3$), Seed and Fertilizer cost($J_4$). Then, at that time, they sort the kinds of rice which one to decide: Basmati Rice($R_1$), Jasmine Rice($R_2$), Long-Grain White Rice($R_3$), Short-Grain Sushi Rice($R_4$). The aforesaid $MADM$ problem is solved in the $IVPF$ environment by using the $IVPFSSPWA$ and $IVPFSSPWG$ operators.

**Step 1:** Data collection in the matrix form in Table 1.

**Step 2:** Normalize the data according to the above-proposed technique in Table 2.

**Step 3:** Using normalized Hamming distances, compute the supports of $\tilde{\sigma}_{kt}$ from $\tilde{\sigma}_{kr}$ for each $k = 1, 2, 3, 4$, where $t, l = 1, 2, 3, 4$ and $t \neq r$.

First, let $k = 1$.

**Table 1. Decision-matrix of interval-valued pythagorean fuzzy set taken by $\tilde{U}$.**

|       | $J_1$ | $J_2$ | $J_3$ | $J_4$ |
|-------|-------|-------|-------|-------|
| $R_1$ | $\langle[0.3, 0.8], [0.4, 0.5]\rangle$ | $\langle[0.2, 0.6], [0.3, 0.5]\rangle$ | $\langle[0.3, 0.6], [0.1, 0.3]\rangle$ | $\langle[0.2, 0.5], [0.4, 0.7]\rangle$ |
| $R_2$ | $\langle[0.3, 0.7], [0.4, 0.5]\rangle$ | $\langle[0.5, 0.7], [0.1, 0.2]\rangle$ | $\langle[0.7, 0.9], [0.2, 0.3]\rangle$ | $\langle[0.3, 0.5], [0.4, 0.8]\rangle$ |
| $R_3$ | $\langle[0.3, 0.6], [0.1, 0.2]\rangle$ | $\langle[0.4, 0.5], [0.1, 0.2]\rangle$ | $\langle[0.2, 0.5], [0.2, 0.3]\rangle$ | $\langle[0.1, 0.3], [0.3, 0.6]\rangle$ |
| $R_4$ | $\langle[0.2, 0.4], [0.1, 0.2]\rangle$ | $\langle[0.5, 0.6], [0.2, 0.3]\rangle$ | $\langle[0.2, 0.4], [0.2, 0.3]\rangle$ | $\langle[0.2, 0.4], [0.2, 0.5]\rangle$ |

**Table 2. Normalized matrix.**

|  | $J_1$ | $J_2$ | $J_3$ | $J_4$ |
|---|---|---|---|---|
| $R_1$ | $\langle[0.3, 0.8], [0.4, 0.5]\rangle$ | $\langle[0.2, 0.6], [0.3, 0.5]\rangle$ | $\langle[0.3, 0.6], [0.1, 0.3]\rangle$ | $\langle[0.4, 0.7], [0.2, 0.5]\rangle$ |
| $R_2$ | $\langle[0.3, 0.7], [0.4, 0.5]\rangle$ | $\langle[0.5, 0.7], [0.1, 0.2]\rangle$ | $\langle[0.7, 0.9], [0.2, 0.3]\rangle$ | $\langle[0.4, 0.8], [0.3, 0.5]\rangle$ |
| $R_3$ | $\langle[0.3, 0.6], [0.1, 0.2]\rangle$ | $\langle[0.4, 0.5], [0.1, 0.2]\rangle$ | $\langle[0.2, 0.5], [0.2, 0.3]\rangle$ | $\langle[0.3, 0.6], [0.1, 0.3]\rangle$ |
| $R_4$ | $\langle[0.2, 0.4], [0.1, 0.2]\rangle$ | $\langle[0.5, 0.6], [0.2, 0.3]\rangle$ | $\langle[0.2, 0.4], [0.2, 0.3]\rangle$ | $\langle[0.2, 0.5], [0.2, 0.4]\rangle$ |

Then supports for $\tilde{\sigma}_{11}$ from $\tilde{\sigma}_{12}, \tilde{\sigma}_{13}, \tilde{\sigma}_{14}$ are computed as follows:

$$\varsigma(\tilde{\sigma}_{11}, \tilde{\sigma}_{12}) = 1 - d(\tilde{\sigma}_{11}, \tilde{\sigma}_{12})$$
$$= 1 - \frac{(|(\mu_{11}^{lb})^2 - (\mu_{12}^{lb})^2| + |(\mu_{11}^{ub})^2 - (\mu_{12}^{ub})^2|) + (|(v_{11}^{lb})^2 - (v_{12}^{lb})^2| + |(v_{11}^{ub})^2 - (v_{12}^{ub})^2|)}{4}$$
$$= 0.9000,$$

$\varsigma(\tilde{\sigma}_{11}, \tilde{\sigma}_{13}) = 0.8525, \varsigma(\tilde{\sigma}_{11}, \tilde{\sigma}_{14}) = 0.9150$, respectively.

Thus, the 3-tupled vector, $\mathbb{S}_{1r}^1(r \neq 1, r = 2, 3, 4) = (0.9000.8525.9150)$ is found.

Similarly, the other tuple vectors for $k = 1$ are:

$$\mathbb{S}_{2r}^1(r = 1, 3, 4) = (0.9000.9275.9250),$$

$$\mathbb{S}_{3r}^1(r = 1, 2, 4) = (0.8525.9275.9025),$$

$$\mathbb{S}_{4r}^1(r = 1, 2, 3) = (0.9150.9250.9025).$$

Similarly, all other tuples $\mathbb{S}_{tr}^k(r \neq t, r = 1, 2, 3, 4)$, for $k = 2, 3, 4$ and $t = 1, 2, 3, 4$ are obtained.

Thus, the support matrix is as follows:

$$= \begin{pmatrix} (0.9000.8525.9150) & (0.9000.9275.9250) & (0.8525.9275.9025) & (0.9150.9250.9025) \\ (0.8700.7500.9275) & (0.8700.8400.8675) & (0.7500.8400.8225) & (0.9275.8675.8225) \\ (0.9550.9400.9875) & (0.9550.9500.9425) & (0.9400.9500.9525) & (0.9875.9425.9525) \\ (0.8775.9800.9400) & (0.8775.8975.9025) & (0.9800.8975.9600) & (0.9400.9025.9600) \end{pmatrix}$$

Now, calculate the sum of supports $T(\tilde{\sigma}_{kt})$ for k = 1,2,3,4 and t = 1,2,3,4 by using the support matrix.

In the case of $k = 1$, the sum of the first alternative's supports across different attributes is as follows:

$T(\tilde{\sigma}_{11}) = \sum_{r=2}^4 \mathbb{S}_{1r}^1 = 2.6675$, and similarly,
$T(\tilde{\sigma}_{12}) = 2.7525, T(\tilde{\sigma}_{13}) = 2.6825, T(\tilde{\sigma}_{14}) = 2.7425$.

Now, for $k = 2, 3, 4$, the equivalent sum of all supports is obtained as

$T(\tilde{\sigma}_{21}) = 2.5475, T(\tilde{\sigma}_{22}) = 2.5775, T(\tilde{\sigma}_{23}) = 2.4125, T(\tilde{\sigma}_{24}) = 2.6175;$

$T(\tilde{\sigma}_{31}) = 2.8825, T(\tilde{\sigma}_{32}) = 2.8475, T(\tilde{\sigma}_{33}) = 2.8425, T(\tilde{\sigma}_{34}) = 2.8825;$

$T(\tilde{\sigma}_{41}) = 2.7975, T(\tilde{\sigma}_{42}) = 2.6775, T(\tilde{\sigma}_{43}) = 2.8375, T(\tilde{\sigma}_{44}) = 2.8025.$

**Step 4:** Calculate the power weights $\lambda_{kt}$, corresponding to $\tilde{\sigma}_{kt}$ by using known weights {0.3000, 0.2000, 0.1000, 0.4000} are:

$$\lambda_{11} = \frac{\omega_1(1 + T(\sigma_{11}))}{\sum_{t=1}^{4} \omega_t(1 + T(\sigma_{1t}))} = 0.2961$$

Similarly, we calculate $\lambda_{kt}$ for $k = 2, 3, 4$ and $t = 1, 2, 3, 4$. Finally, we get the power weight matrix, $(\lambda_{kt})_{4\times4}$, as

$$= \begin{pmatrix} 0.2961 & 0.2020 & 0.0991 & 0.4029 \\ 0.2983 & 0.2005 & 0.0956 & 0.4055 \\ 0.3009 & 0.1988 & 0.0993 & 0.4011 \\ 0.3014 & 0.1946 & 0.1015 & 0.4024 \end{pmatrix}$$

**Step 5:** Utilizing the *IVPFSSPWA* operator (or, *IVPFSSPWG* operator) for $\eta = -2$, aggregate all attribute values corresponding to each alternative to obtain a single *IVPFN* for each alternative.

$$\begin{aligned} \tilde{\sigma}_1 &= IVPFSSPWA(\tilde{\sigma}_{11}, \tilde{\sigma}_{12}, \tilde{\sigma}_{13}, \tilde{\sigma}_{14}) \\ &= \left( \left[ \sqrt{1 - (\sum_{t=1}^{4} \lambda_{1t}(1 - (\mu_{1t}^{lb})^2)^\eta)^{1/\eta}}, \sqrt{1 - (\sum_{t=1}^{4} \lambda_{1t}(1 - (\mu_{1t}^{ub})^2)^\eta)^{1/\eta}} \right] \right. \\ &\quad \left. , \left[ \sqrt{(\sum_{t=1}^{4} \lambda_{1t}(v_{1t}^{lb})^{2\eta})^{1/\eta}}, \sqrt{(\sum_{t=1}^{4} \lambda_{1t}(v_{1t}^{ub})^{2\eta})^{1/\eta}} \right] \right) \\ &= ([0.3343, 0.7295], [0.1672, 0.4401]). \end{aligned}$$

Similarly,
$\tilde{\sigma}_2 = ([0.4729, 0.7997], [0.1473, 0.2865]), \tilde{\sigma}_3 = ([0.3177, 0.5766], [0.1025, 0.2274]),$
$\tilde{\sigma}_4 = ([0.3075, 0.4965], [0.1305, 0.2539]).$

**Step 6:** From (3), the score value of the concluding *IVPFNs* $\tilde{\sigma}_k (k = 1, 2, 3, 4)$ as $\wp(\tilde{\sigma}_1) = 0.2111, \wp(\tilde{\sigma}_2) = 0.3797, \wp(\tilde{\sigma}_3) = 0.1856, \wp(\tilde{\sigma}_4) = 1298.$

**Step 7:** Utilizing score values, rank the alternatives, and the result is as follows:

$$R_2 > R_1 > R_3 > R_4.$$

The best option is determined to be $R_2$ since it has the highest score value among the alternatives.

Or,

$$\begin{aligned} \tilde{\sigma}_1 &= IVPFSSPWG(\tilde{\sigma}_{11}, \tilde{\sigma}_{12}, \tilde{\sigma}_{13}, \tilde{\sigma}_{14}) \\ &\left( \left[ \sqrt{(\sum_{t=1}^{4} \lambda_{1t}(\mu_{1t}^{lb})^{2\eta})^{1/\eta}}, \sqrt{(\sum_{t=1}^{4} \lambda_{1t}(\mu_{1t}^{ub})^{2\eta})^{1/\eta}} \right] \right. \\ &\quad \left. , \left[ \sqrt{1 - (\sum_{t=1}^{4} \lambda_{1t}(1 - (v_{1t}^{lb})^2)^\eta)^{1/\eta}}, \sqrt{1 - (\sum_{t=1}^{4} \lambda_{1t}(1 - (v_{1t}^{ub})^2)^\eta)^{1/\eta}} \right] \right) \\ &= ([0.2691, 0.6781], [0.2964, 0.4877]). \end{aligned}$$

Similarly,
$\tilde{\sigma}_2 = ([0.3651, 0.7469], [0.3081, 0.4533]), \tilde{\sigma}_3 = ([0.2827, 0.5598], [0.1147, 0.2570]),$
$\tilde{\sigma}_4 = ([0.2108, 0.4533], [0.1766, 0.3276]).$

**Step 6':** From (3), the score value of the concluding *IVPFNs* $\tilde{\sigma}_k (k = 1, 2, 3, 4)$ as $\wp(\tilde{\sigma}_1) = 0.1033, \wp(\tilde{\sigma}_2) = 0.1954, \wp(\tilde{\sigma}_3) = 0.1570, \wp(\tilde{\sigma}_4) = 0.0557.$

**Step 7':** Utilizing score values, rank the alternatives, and the result is as follows:

$$R_2 > R_3 > R_1 > R_4.$$

The best option is determined to be $R_2$ since it has the highest score value among the alternatives.

## 6 Sensitivity analysis

Analyzing the impact of changes in input factors on track factors involves the utilization of a financial model known as sensitivity analysis. This approach forecasts decision outcomes based on crucial variables, specifically focusing on the sensitivity of a parameter (denoted as $\eta$ < 0) and sensitivity related to different weights. The IVPFSSPWA operator and IVPFSSPWG operator generate two sequences with slight variations in alternatives, yet the optimal alternative $R_2$ remains consistent. Sensitivity analysis plays a vital role in handling uncertainty in mathematical models when input values fluctuate. It is commonly employed alongside uncertainty analysis to improve the precision of research and model constructions, which depend on assumptions about input accuracy. The application of sensitivity analysis assists in computation, prediction, and the identification of areas requiring cycle enhancements or modifications. However, it's crucial to recognize that using historical data may occasionally lead to inaccurate projections, as past events do not necessarily predict future ones.

### 6.1 "$\eta$" parameter sensitivity

In this portion, we simply perform the responsiveness analysis utilizing the $\eta$-IVPFSSPWA operator and $\eta$-IVPFSSPWG operator in Tables 3 and 4, respectively, to examine the effects of varying $\eta < 0$ parameter values on the ranking of the other options. And the Figs 2 and 3 show that when we decrease the values of $\eta$, very nothing changes. Additionally, we have observed that when the values of $\eta$ shrink, the values of the score function of each choice become more modest and the best alternative, $R_2$, remains unaltered.

**Table 3. A different ranking by altering the parameter values using IVPFSSPWA operator.**

| $\eta$-values: | Values of score function | Ranking |
|---|---|---|
| $\eta = -2$ | $\wp(\tilde{\sigma}_1) = 0.2111, \wp(\tilde{\sigma}_2) = 0.3797, \wp(\tilde{\sigma}_3) = 0.1856, \wp(\tilde{\sigma}_4) = 0.1298$ | $R_2 > R_1 > R_3 > R_4$ |
| $\eta = -4$ | $\wp(\tilde{\sigma}_1) = 0.2480, \wp(\tilde{\sigma}_2) = 0.4355, \wp(\tilde{\sigma}_3) = 0.1902, \wp(\tilde{\sigma}_4) = 0.1468$ | $R_2 > R_1 > R_3 > R_4$ |
| $\eta = -8$ | $\wp(\tilde{\sigma}_1) = 0.2853, \wp(\tilde{\sigma}_2) = 0.5084, \wp(\tilde{\sigma}_3) = 0.1960, \wp(\tilde{\sigma}_4) = 0.1718$ | $R_2 > R_1 > R_3 > R_4$ |
| $\eta = -10$ | $\wp(\tilde{\sigma}_1) = 0.2950, \wp(\tilde{\sigma}_2) = 0.5306, \wp(\tilde{\sigma}_3) = 0.1981, \wp(\tilde{\sigma}_4) = 0.1827$ | $R_2 > R_1 > R_3 > R_4$ |

**Table 4. A different ranking by altering the parameter values using IVPFSSPWG operator.**

| $\eta$-values: | Values of score function | Ranking |
|---|---|---|
| $\eta = -2$ | $\wp(\tilde{\sigma}_1) = 0.1033, \wp(\tilde{\sigma}_2) = 0.1954, \wp(\tilde{\sigma}_3) = 0.1570, \wp(\tilde{\sigma}_4) = 0.0557$ | $R_2 > R_3 > R_1 > R_4$ |
| $\eta = -4$ | $\wp(\tilde{\sigma}_1) = 0.0837, \wp(\tilde{\sigma}_2) = 0.1763, \wp(\tilde{\sigma}_3) = 0.1452, \wp(\tilde{\sigma}_4) = 0.0466$ | $R_2 > R_3 > R_1 > R_4$ |
| $\eta = -8$ | $\wp(\tilde{\sigma}_1) = 0.0602, \wp(\tilde{\sigma}_2) = 0.1528, \wp(\tilde{\sigma}_3) = 0.1292, \wp(\tilde{\sigma}_4) = 0.0362$ | $R_2 > R_3 > R_1 > R_4$ |
| $\eta = -10$ | $\wp(\tilde{\sigma}_1) = 0.0525, \wp(\tilde{\sigma}_2) = 0.1450, \wp(\tilde{\sigma}_3) = 0.1244, \wp(\tilde{\sigma}_4) = 0.0329$ | $R_2 > R_3 > R_1 > R_4$ |

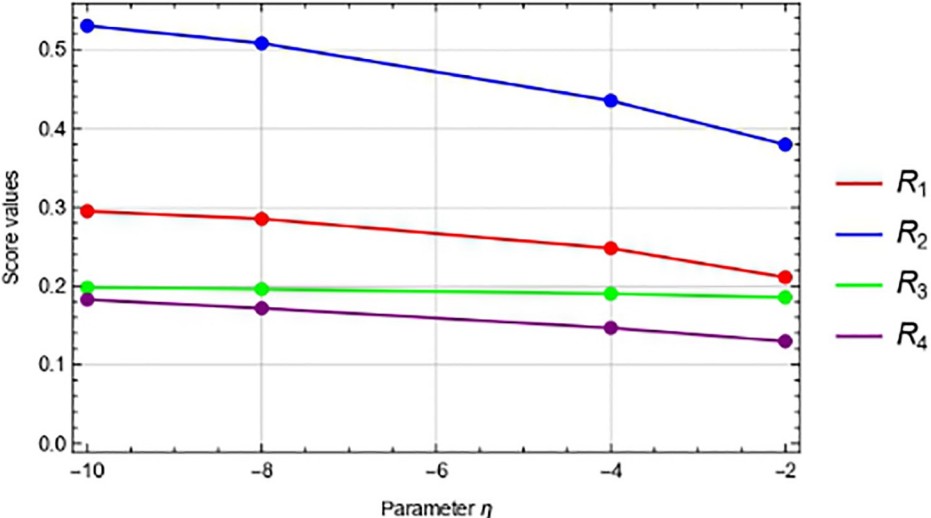

**Fig 2. Using IVPFSSPWA operator, the graphical representation of score values for $\eta$.**

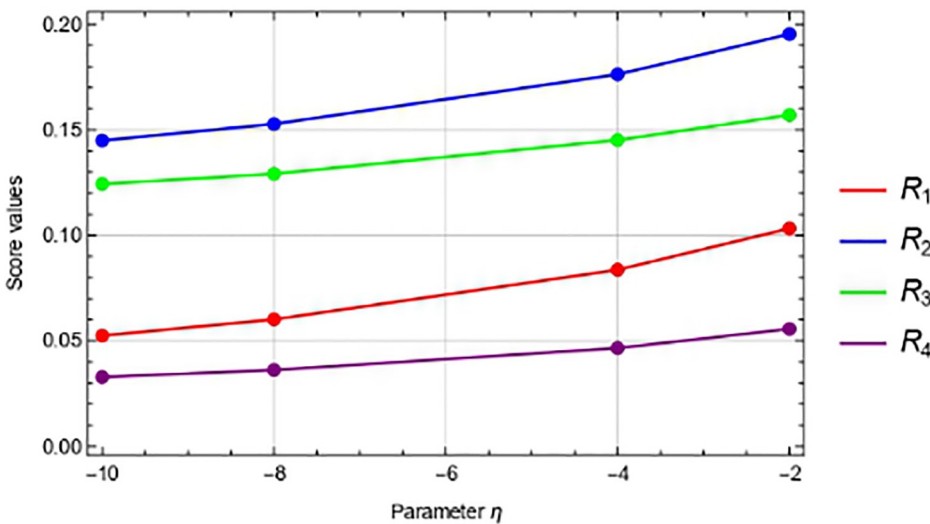

**Fig 3. Using IVPFSSPWG operator, the graphical representation of score values for $\eta$.**

## 6.2 Sensitivity analysis w.r.t. weights of characteristics

An approach used in decision-making processes, particularly in MADM, is sensitivity analysis concerning attribute weights, which evaluates the consequences of different weightings assigned to the qualities under evaluation as depicted in Tables 5 and 6. In these scenarios of decision-making, several features or criteria are often evaluated, and these attributes may be assigned different weights or degrees of relevance. These weights are often applied based on the preferences of the decision-maker, domain knowledge, or other factors. Sensitivity analysis involves carefully adjusting the weights assigned to each attribute and seeing how those adjustments impact the judgment or conclusion that is reached at the end. The goals of this analysis

**Table 5. Sensitivity analysis w.r.t. weights of characteristics using IVPFSSPWA operator.**

| Values of new Weights: | Values of score function | Ranking |
|---|---|---|
| {0.3000, 0.2000, 0.1000, 0.4000} | $\wp(\tilde{\sigma}_1) = 0.2111, \wp(\tilde{\sigma}_2) = 0.3797, \wp(\tilde{\sigma}_3) = 0.1856, \wp(\tilde{\sigma}_4) = 0.1298$ | $R_2 > R_1 > R_3 > R_4$ |
| {0.4000, 0.2500, 0.1250, 0.2250} | $\wp(\tilde{\sigma}_1) = 0.2155, \wp(\tilde{\sigma}_2) = 0.3900, \wp(\tilde{\sigma}_3) = 0.1854, \wp(\tilde{\sigma}_4) = 0.1392$ | $R_2 > R_1 > R_3 > R_4$ |
| {0.3940, 0.2963, 0.2173, 0.0924} | $\wp(\tilde{\sigma}_1) = 0.2190, \wp(\tilde{\sigma}_2) = 0.4349, \wp(\tilde{\sigma}_3) = 0.1780, \wp(\tilde{\sigma}_4) = 0.1449$ | $R_2 > R_1 > R_3 > R_4$ |
| {0.3096, 0.2345, 0.1973, 0.2586} | $\wp(\tilde{\sigma}_1) = 0.2191, \wp(\tilde{\sigma}_2) = 0.4268, \wp(\tilde{\sigma}_3) = 0.1785, \wp(\tilde{\sigma}_4) = 0.1335$ | $R_2 > R_1 > R_3 > R_4$ |

**Table 6. Sensitivity analysis w.r.t. weights of characteristics using IVPFSSPWG operator.**

| Values of new Weights: | Values of score function | Ranking |
|---|---|---|
| {0.3000, 0.2000, 0.1000, 0.4000} | $\wp(\tilde{\sigma}_1) = 0.1033, \wp(\tilde{\sigma}_2) = 0.1954, \wp(\tilde{\sigma}_3) = 0.1570, \wp(\tilde{\sigma}_4) = 0.0557$ | $R_2 > R_3 > R_1 > R_4$ |
| {0.4000, 0.2500, 0.1250, 0.2250} | $\wp(\tilde{\sigma}_1) = 0.0924, \wp(\tilde{\sigma}_2) = 0.1867, \wp(\tilde{\sigma}_3) = 0.1549, \wp(\tilde{\sigma}_4) = 0.0618$ | $R_2 > R_3 > R_1 > R_4$ |
| {0.3940, 0.2963, 0.2173, 0.0924} | $\wp(\tilde{\sigma}_1) = 0.0855, \wp(\tilde{\sigma}_2) = 0.2048, \wp(\tilde{\sigma}_3) = 0.1429, \wp(\tilde{\sigma}_4) = 0.0647$ | $R_2 > R_3 > R_1 > R_4$ |
| {0.3096, 0.2345, 0.1973, 0.2586} | $\wp(\tilde{\sigma}_1) = 0.0967, \wp(\tilde{\sigma}_2) = 0.2108, \wp(\tilde{\sigma}_3) = 0.1441, \wp(\tilde{\sigma}_4) = 0.0574$ | $R_2 > R_3 > R_1 > R_4$ |

are to identify the attributes that have the greatest influence on the decision and the degree to which the decision or outcome is affected by the relative weights assigned to each characteristic. Sensitivity analysis would entail adjusting the weights assigned to each criterion and monitoring shifts in the relative rankings of various rice varieties in the agricultural sector. This study may be used by farmers, researchers, and policymakers to determine which types of rice are most crucial to their decision and adjust the weights of those variables accordingly. It's a useful tool for examining decision-making problems and assessing how adaptable decisions are to shifts in the relative weight of different attributes.

By assigning the alternate weights, it is clear from Figs 4 and 5 that the ranking of rice is a little bit of a change and the best alternative, $R_2$, remains unaltered.

## 7 Comparison analysis

To further demonstrate the merits and benefits of the suggested methods, we lead the subsequent comparability likeness. To compare our suggested operators with other existing ones, we find that our proposed operators are more versatile than their counterparts. In comparison to the IVIFSSPA and IVIFSSPG operators introduced by Zindani *et al.* [26], our operators yield the same optimal alternatives as shown in the Table 7. However, the IVIFS environment has limitations when the sum of BG and NG exceeds one. This is why our aggregation operators, coupled with our IVPFS environment, prove to be more generalized and compatible.

When comparing our operators with the IPFWA and IPFWG operators presented by Harish Garg [32], the resulting optimal alternatives align with the Table 7. Although the environments are the same, our aggregation operators based on SS operations, offer flexibility through the parameter $\eta < 0$. Consequently, our proposed technique is more suitable for solving decision-making problems compared to other existing measures. Therefore, our approach effectively addresses decision-making problems under the IVPFS framework.

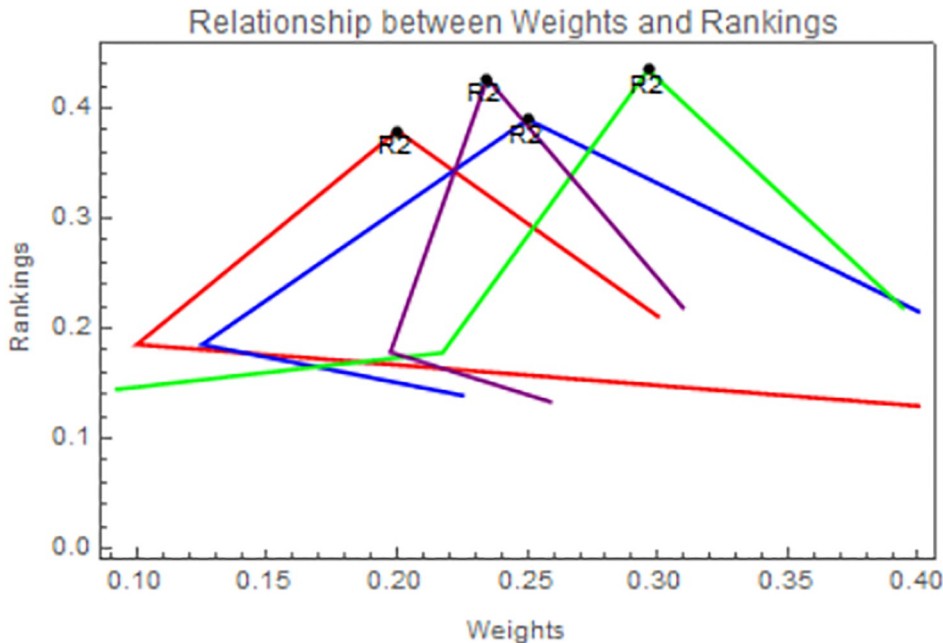

**Fig 4. Using IVPFSSPWA operator, relation between weights and ranking.**

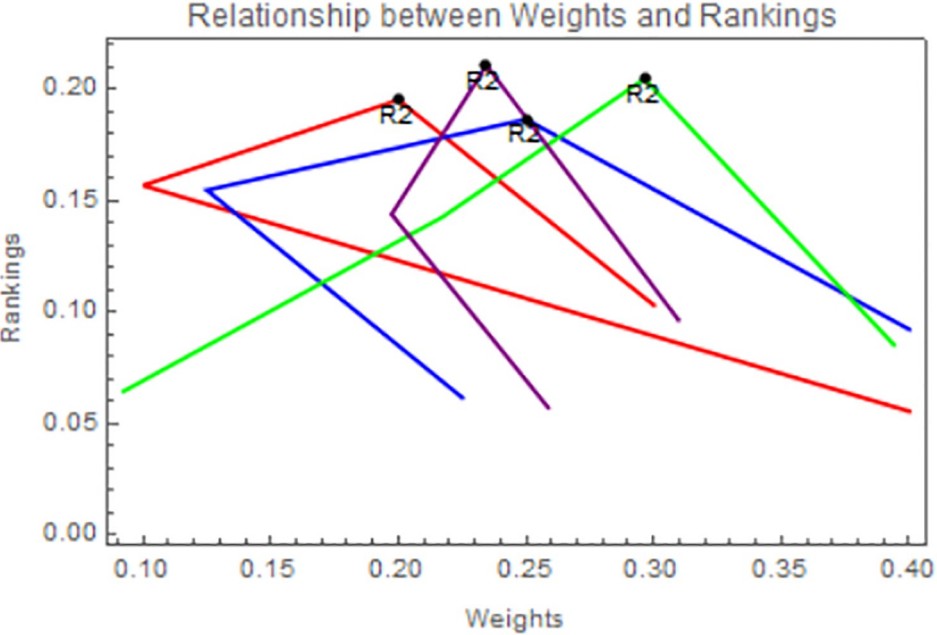

**Fig 5. Using IVPFSSPWG operator, relation between weights and ranking.**

Such methodologies have clear limitations while becoming uncertain, most importantly, when applied in dynamic fuzzy scenarios. Their propensity to cause a decrease in performance on linking themselves with big data sets bounds their application in routine life situations that interact with large data processing. The way to select the optimal parameters for these

**Table 7. Comparative methods.**

| Methods | Values of score function | Ranking |
|---|---|---|
| IVIFSSPWA operator [26] | $\wp(\tilde{\sigma}_1) = 0.1948, \wp(\tilde{\sigma}_2) = 0.3590, \wp(\tilde{\sigma}_3) = 0.1832,$ $\wp(\tilde{\sigma}_4) = 0.1201$ | $R_2 > R_1 > R_3 > R_4$ |
| IVIFSSPWG operator [26] | $\wp(\tilde{\sigma}_1) = 0.1166, \wp(\tilde{\sigma}_2) = 0.2080, \wp(\tilde{\sigma}_3) = 0.1637,$ $\wp(\tilde{\sigma}_4) = 0.0633$ | $R_2 > R_3 > R_1 > R_4$ |
| IPFWA [32] | $\wp(\tilde{\sigma}_1) = 0.1645, \wp(\tilde{\sigma}_2) = 0.2873, \wp(\tilde{\sigma}_3) = 0.1786,$ $\wp(\tilde{\sigma}_4) = 0.1043$ | $R_2 > R_3 > R_1 > R_4$ |
| IPFWG [32] | $\wp(\tilde{\sigma}_1) = 0.1299, \wp(\tilde{\sigma}_2) = 0.2240, \wp(\tilde{\sigma}_3) = 0.1688,$ $\wp(\tilde{\sigma}_4) = 0.0729$ | $R_2 > R_3 > R_1 > R_4$ |
| IVPFSSPWA (proposed operator) | $\wp(\tilde{\sigma}_1) = 0.2111, \wp(\tilde{\sigma}_2) = 0.3797, \wp(\tilde{\sigma}_3) = 0.1856,$ $\wp(\tilde{\sigma}_4) = 0.1298$ | $R_2 > R_1 > R_3 > R_4$ |
| IVPFSSPWG (proposed operator) | $\wp(\tilde{\sigma}_1) = 0.1033, \wp(\tilde{\sigma}_2) = 0.1954, \wp(\tilde{\sigma}_3) = 0.1570,$ $\wp(\tilde{\sigma}_4) = 0.0557$ | $R_2 > R_3 > R_1 > R_4$ |

operators is unvaried and entails experiencing, the inclusion of a subjective component that may hinder their usage in everyday situations. Moreover, these operators allow a gateway to an advanced structure for creating choices without worrying about ambiguity. To calculate their efficacy in different kinds of circumstances wisely is very significant. This is because they might not precisely coincide with the choices of the different decision-makers.

## 7.1 Validation test

The purpose of the suggested operators, IVPFSSPWA and IVPFSSPWG, is to enhance the decision-making procedure by using advanced score functions. Table 7 findings and Table 8 Spearman's rank correlation coefficients show this:

- An ideal +ve correlation (1.0) between the operator IVPFSSPWA and the operator IVIFSSPWA indicates that their ranks are the same. Along with the IPFWG, IPFWA, and IVIFSSPWG operators, there are modest correlations (0.4), suggesting a certain degree of commonality and a few differences among the ranks.

- The IPFWG, IPFWA, and IVIFSSPWG operators have an ideal +ve correlation (1.0) with the operator IVPFSSPWG, suggesting that their ranks are similar. Additionally, it has an average correlation (0.4) with the operators IVIFSSPWA and IVPFSSPWA, indicating some variations in the ranks generated.

The suggested operators execute well in general with a significant correlation compared to the existing operators, proving their usefulness in the decision-making procedure. Their constancy and dependability are demonstrated by the flawless correlations among different operators.

**Table 8. Spearman's rank correlation coefficient for comparisons.**

| | IVIFSSPWA | IVIFSSPWG | IPFWA | IPFWG | IVPFSSPWA | IVPFSSPWG |
|---|---|---|---|---|---|---|
| **IVIFSSPWA** | 1.0 | 0.4 | 0.4 | 0.4 | 1.0 | 0.4 |
| **IVIFSSPWG** | 0.4 | 1.0 | 1.0 | 1.0 | 0.4 | 1.0 |
| **IPFWA** | 0.4 | 1.0 | 1.0 | 1.0 | 0.4 | 1.0 |
| **IPFWG** | 0.4 | 1.0 | 1.0 | 1.0 | 0.4 | 1.0 |
| **IVPFSSPWA** | 1.0 | 0.4 | 0.4 | 0.4 | 1.0 | 0.4 |
| **IVPFSSPWG** | 0.4 | 1.0 | 1.0 | 1.0 | 0.4 | 1.0 |

## 8 Conclusions

The primary contribution of this article is the provision of several effective IVPF aggregate operators, specifically the IVPFSSPA, IVPFSSPG, IVPFSSPWA, and IVPFSSPWG operators. These operators increase the adaptability of the information fusion process when there are SS operations associated with each of them. The ability of the generated operators to make accurate judgments by mitigating the impact of erroneous information supplied by the affected decision-maker through the use of a power aggregation operator is among their most essential characteristics. Using the proposed operators, a novel approach to handling MADM with IVPF information has been presented.

The optimal rice quality selection decision-making problem is addressed by the developed method. Moreover, the impact of the SS component on the ranking results has been presented to illustrate the benefits of the current methodology. Finally, the efficiency and sensitivity of the suggested operators in handling situations requiring decision-making have been demonstrated by comparison with alternative methods.

The established operators could be extended to more ambiguous domains in the future, such as Interval-valued dual hesitant IVDH, q-rung IV fuzzy, quasirung fuzzy sets [33, 34], and others. However, it is believed that the suggested strategy would clear the way for new approaches to handling decision-making problems in varied fuzzy contexts. Several MADM algorithms such as WASPAS, ELECTRE, and TODIM may benefit from the increased effectiveness of using the intended aggregation operators.

## Supporting information

**S1 Dataset. Minimal data set file.**
(PDF)

**S1 File.**
(PDF)

**S2 File.**
(PDF)

## Acknowledgments

The authors would like to express appreciation to the anonymous reviewers and Editors for their very helpful comments that improved the paper.

## Author Contributions

**Funding acquisition:** Ying Wang.

**Investigation:** Usman Khalid.

**Methodology:** Jawad Ali.

**Project administration:** Muhammad Ahsan Binyamin.

**Resources:** Usman Khalid, Muhammad Ahsan Binyamin.

**Software:** Usman Khalid, Jawad Ali.

**Supervision:** Muhammad Ahsan Binyamin.

**Validation:** Jawad Ali.

**Visualization:** Jawad Ali.

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
