## [Decision Letter · Decision Letter 0]

1 Jul 2024

PONE-D-24-07498A Novel Approach to Decision Making in Rice Quality Management using Interval-Valued Pythagorean fuzzy Schweizer and Sklar Power Aggregation OperatorsPLOS ONE

Dear Dr. Binyamin,

Thank you for submitting your manuscript to PLOS ONE. After careful consideration, we feel that it has merit but does not fully meet PLOS ONE’s publication criteria as it currently stands. Therefore, we invite you to submit a revised version of the manuscript that addresses the points raised during the review process.

We look forward to receiving your revised manuscript.

Kind regards,

Naeem Jan, PhD 

Academic Editor

PLOS ONE

“No”

Additional Editor Comments:

Dear Authors.

Please revise your paper based on the reviewers comments.

Thank you

Reviewers' comments:

Reviewer's Responses to Questions

**Comments to the Author**

1. Is the manuscript technically sound, and do the data support the conclusions?

Reviewer #1: Yes

Reviewer #2: Partly

2. Has the statistical analysis been performed appropriately and rigorously? 

Reviewer #1: N/A

Reviewer #2: N/A

3. Have the authors made all data underlying the findings in their manuscript fully available?

Reviewer #1: Yes

Reviewer #2: Yes

4. Is the manuscript presented in an intelligible fashion and written in standard English?

Reviewer #1: No

Reviewer #2: No

5. Review Comments to the Author

Reviewer #1: The authors introduce the Schweizer and Sklar power geometric operator for the interval-valued Pythagorean fuzzy set. The presented idea is interesting and suitable for this journal. But, I have some necessary suggestions for authors given as follows:

1. The abstract must be more precise and focus on the objective: Why is the study needed?

2. Brief results must be shown in the Abstract, and at present, it is missing.

3. Literature review - Please improve on this, and state clearly what the focus of the paper is and why the focus is presented. What importance it brings to the study line is not clear.

4. There is a clear lack of motivation as to why the authors chose these aggregation operators in an interval-valued Pythagorean fuzzy environment.

5. Please bring some facts and figures in the introduction to support the ideas about neutrosophic sets and other fuzzy sets such as: (a): Comparison between fuzzy soft matrix (fsm) and interval valued fuzzy soft matrix (ivfsm) in decision making. (b): Aggregation Operators for Interval-Valued Pythagorean Fuzzy Soft Set with Their Application to Solve Multi-Attribute Group Decision Making Problem. (c): Einstein aggregation operators for Pythagorean fuzzy soft sets with their application in multiattribute group decision-making. (d): An integrated group decision-making technique under interval-valued probabilistic linguistic T-spherical fuzzy information and its application to the selection of cloud storage. (e): An interaction and feedback mechanism-based group decision-making for emergency medical supplies supplier selection using T-spherical fuzzy information. (f): An Intelligent MCGDM Model in Green Suppliers Selection Using Interactional Aggregation Operators for Interval-Valued Pythagorean Fuzzy Soft Sets. (g): Extension of correlation coefficient based TOPSIS technique for interval-valued Pythagorean fuzzy soft set: A case study in extract, transform, and load techniques. (h): A Decision-Making Approach Based on Score Matrix for Pythagorean Fuzzy Soft Set.

6. There are many symbols, so please explain all of them clearly to help readers understand.

7. Missing the definitions of fuzzy sets and Pythagorean fuzzy sets in the preliminaries section.

8. What is the source of the data used in section 5?

9. Provide a better validation section with comparisons with the existing models. Add more discussion on the results and discuss advantages and limitations.

10. Comparison can be made effective, too, with more results and discussion and also expand the experimental setup.

11. The authors need to explain the contribution and objectives of this research in a wide manner.

12. Findings, limitations, and recommendations of this paper can be discussed more in the conclusion section.

13. Please bring and focus on future research directions.

14. Some sentences in the text need to be rewritten. The manuscript needs to be carefully edited by professional English editors, paying special attention to English grammar, spelling, and sentence structure so that readers can clearly understand the objectives and results of the research.

Reviewer #2: The present paper proposes a series of interval-valued Pythagorean fuzzy Schweizer and Sklar power aggregation operators. The operators are then utilized to MADM problem. The topic is interesting and relevant. Overall, the paper is well-written and organized, and the methodology is adequately described. However, some areas require further clarification and improvement before it can be considered for publication.

1. The introductory section must appropriately highlight the motivations and objectives of this research by reporting what are the performance improvements compared to well-known fuzzy multi-criteria decision-making in the literature.

2. Clearly articulate why “interval-valued Pythagorean fuzzy " was chosen as the research topic and highlight its practical relevance. What are the advantages?

3. There are too many redundant references in the introduction, some of which can be deleted. There are citations that comprise of multiple references for eg. ([6];[7];[8];[9]), ([10];[11];[12];[13]), ([23];[24];[25]). It is advised to refer to each reference individually in the text rather than grouping them together.

4. The main contribution of the manuscript should be listed in bullets under the Introduction section.

5. Several aggregation operators exist in the literature, such as Dombi, Archimedean, Frank etc. why authors select the Schweizer and Sklar power aggregation operators is not clear. Please refer to and cite properly to demonstrate the superiority of your work. q-Rung Orthopair Fuzzy Archimedean Aggregation Operators: Application in the Site Selection for Software Operating Units; Intuitionistic fuzzy Dombi aggregation operators and their application to multiple attribute decision-making; Q-rung orthopair fuzzy Frank aggregation operators and its application in multiple attribute decision-making with unknown attribute weights. What are the advantages?

6. In Step 6 and 7 of Algorithm of Section 4 if score values become equal what will happen? Modify the algorithm with the use of accuracy function.

7. Remove background fill of Fig. 1.

8. In section 5 Step 1 is written as “Data collection in the matrix form (For q=3) in table [1]”. Why q=3?

9. In Tables 5 & 6 of Sensitivity analysis how new sets of weights are generated? Mention.

10. Comparative analysis needs to be stronger. I hope the author can add more comparisons to prove the rationality and effectiveness of the proposed method. Table 7 shows that the ranking obtained by the proposed method and the comparative method are identical. Therefore, it is imperative to ponder why decision-makers should choose this method over existing ones.

11. Your method could be applied to another extension of fuzzy sets such as ‘quasirung fuzzy sets‟. Include these in the future scope of the conclusion section by properly citing the following references. https://doi.org/10.1016/j.engappai.2022.105299;
https://doi.org/10.1007/s41066-021-00308-9

12. English writing should be improved considerably throughout the manuscript.

Based on the above comments, I recommend a major revision be made before reconsidering the manuscript.

6. PLOS authors have the option to publish the peer review history of their article (what does this mean?). If published, this will include your full peer review and any attached files.

Reviewer #1: No

Reviewer #2: **Yes: **Mijanur Rahaman Seikh

---

## [Author Response · Author response to Decision Letter 0]

8 Aug 2024

Reviewer#1, Concern # 1: The abstract must be more precise and focus on the objective: Why is the study needed?

Author response: Thank you for your valuable suggestion. 

Author action: We have described all those points highlighted by you along with focusing on main objective behind this study.

Reviewer#1, Concern # 2: Brief results must be shown in the Abstract, and at present, it is missing.

Author response: Thank you so much for your valuable suggestion.

Author action: We have shown the main results in the Abstract.

Reviewer#1, Concern # 3: Literature review - Please improve on this, and state clearly what the focus of the paper is and why the focus is presented. What importance it brings to the study line is not clear.

Author response: Thank you for your suggestion.

Author action: We have improved our literature review and also described motivations about my environment and proposed oprerators.

Reviewer#1, Concern # 4: There is a clear lack of motivation as to why the authors chose these aggregation operators in an interval-valued Pythagorean fuzzy environment.

Author response: Thank you for your valuable suggestion.

Author action: We have already elaborated this in the motivation subsection.

Reviewer#1, Concern # 5: Please bring some facts and figures in the introduction to support the ideas about neutrosophic sets and other fuzzy sets such as: (a): Comparison between fuzzy soft matrix (fsm) and interval valued fuzzy soft matrix (ivfsm) in decision making. (b): Aggregation Operators for Interval-Valued Pythagorean Fuzzy Soft Set with Their Application to Solve Multi-Attribute Group Decision Making Problem. (c): Einstein aggregation operators for Pythagorean fuzzy soft sets with their application in multiattribute group decision-making. (d): An integrated group decision-making technique under interval-valued probabilistic linguistic T-spherical fuzzy information and its application to the selection of cloud storage. (e): An interaction and feedback mechanism-based group decision-making for emergency medical supplies supplier selection using T-spherical fuzzy information. (f): An Intelligent MCGDM Model in Green Suppliers Selection Using Interactional Aggregation Operators for Interval-Valued Pythagorean Fuzzy Soft Sets. (g): Extension of correlation coefficient based TOPSIS technique for interval-valued Pythagorean fuzzy soft set: A case study in extract, transform, and load techniques. (h): A Decision-Making Approach Based on Score Matrix for Pythagorean Fuzzy Soft Set.

Author response: Thank you for the recommendation. 

Author action: We have done it by adding all the above citations.

Reviewer#1, Concern # 6: There are many symbols, so please explain all of them clearly to help readers understand.

Author response: Thank you for suggestion.

Author action: We have tried to reduce the abbreviations in this manuscript.

Reviewer#1, Concern # 7: Missing the definitions of fuzzy sets and Pythagorean fuzzy sets in the preliminaries section.

Author response: Thank you for your valuable suggestion. 

Author action: We have added them in the preliminaries section.

Reviewer#1, Concern # 8: What is the source of the data used in section 5?

Author response: Thank you for asking. 

Author action: We have used rough data instead of real data in section 5.

Reviewer#1, Concern # 9: Provide a better validation section with comparisons with the existing models. Add more discussion on the results and discuss advantages and limitations.

Author response: Thank you for your valuable suggestion and careful review of the article. 

Author action: We have applied validation test in the comparison section that are shown the advantages of our operators and also discussed limitations.

Reviewer#1, Concern # 10: Comparison can be made effective, too, with more results and discussion and also expand the experimental setup.

Author response: Thank you for your valuable suggestion and careful review of the article. 

Author action: We have used validation to create our comparison effectively.

Reviewer#1, Concern # 11: The authors need to explain the contribution and objectives of this research in a wide manner.

Author response: Thank you for your valuable suggestion. 

Author action: We have tried our best to explain them (see in author’s list).

Reviewer#1, Concern # 12: Findings, limitations, and recommendations of this paper can be discussed more in the conclusion section.

Author response: Thank you for your valuable suggestion. 

Author action: We have mentioned limitations above the conclusion section and have tried to discuss more in the conclusion section.

Reviewer#1, Concern # 13: Please bring and focus on future research directions.

Author response: Thank you for your valuable suggestion. 

Author action: We have added the future directions clearly.

Reviewer#1, Concern # 14: Some sentences in the text need to be rewritten. The manuscript needs to be carefully edited by professional English editors, paying special attention to English grammar, spelling, and sentence structure so that readers can clearly understand the objectives and results of the research.

Author response: Thank you for your valuable suggestion. 

Author action: We have tried to rectified them.

Reviewer#2, Concern # 1: The introductory section must appropriately highlight the motivations and objectives of this research by reporting what are the performance improvements compared to well-known fuzzy multi-criteria decision-making in the literature..

Author response: Thank you for your valuable suggestion.

Author action: We have mentioned motivation in the introduction section and also have improved literature.

Reviewer#2, Concern # 2: Clearly articulate why “interval-valued Pythagorean fuzzy " was chosen as the research topic and highlight its practical relevance. What are the advantages?

Author response: Thank you for the suggestion.

Author action: Because this environment is an extension of Pythagorean fuzzy set and interval valued fuzzy set and have described its importance in the motivation subsection.

Reviewer#2, Concern # 3: There are too many redundant references in the introduction, some of which can be deleted. There are citations that comprise of multiple references for eg. ([6];[7];[8];[9]), ([10];[11];[12];[13]), ([23];[24];[25]). It is advised to refer to each reference individually in the text rather than grouping them together.

Author response: Thank you for the suggestion.

Author action: We have successfully removed all bulk references in the introduction section.

Reviewer#2, Concern # 4: The main contribution of the manuscript should be listed in bullets under the Introduction section.

Author response: Thank you so much for your valuable suggestions.

Author action: We have done it accordingly.

Reviewer#3, Concern # 5 Several aggregation operators exist in the literature, such as Dombi, Archimedean, Frank etc. why authors select the Schweizer and Sklar power aggregation operators is not clear. Please refer to and cite properly to demonstrate the superiority of your work. q-Rung Orthopair Fuzzy Archimedean Aggregation Operators: Application in the Site Selection for Software Operating Units; Intuitionistic fuzzy Dombi aggregation operators and their application to multiple attribute decision-making; Q-rung orthopair fuzzy Frank aggregation operators and its application in multiple attribute decision-making with unknown attribute weights. What are the advantages?

Author response: Thank you so much for your suggestion.

Author action: We have mentioned the advantages of our operators using SS norms in the motivation section and also have mentioned all above citations.

Reviewer#2,Concern#6: In Step 6 and 7 of Algorithm of Section 4 if score values become equal what will happen? Modify the algorithm with the use of accuracy function.

Author response: Thank you for suggestion. 

Author action: When the score values are same then we use the accuracy function for ranking system. We have modified our algorithm according to it.

Reviewer#2,Concern#7: Remove background fill of Fig. 1..

Author response: Thank you for suggestion. 

Author action: We have removed it.

Reviewer#2,Concern#8: In section 5 Step 1 is written as “Data collection in the matrix form (For q=3) in table [1]”. Why q=3?

Author response: Thank you for highlighting.

Author action: This was the typo error and we have made it correct by removing it.

Reviewer#2,Concern#9: In Tables 5 & 6 of Sensitivity analysis how new sets of weights are generated? Mention.

Author response: Thank you for asking.

Author action: Basically, here we have used known weight vectors whose sum is exactly equal to one and have also mentioned in our algorithm.

Reviewer#2,Concern#10: Comparative analysis needs to be stronger. I hope the author can add more comparisons to prove the rationality and effectiveness of the proposed method. Table 7 shows that the ranking obtained by the proposed method and the comparative method are identical. Therefore, it is imperative to ponder why decision-makers should choose this method over existing ones.

Author response: Thank you for suggestion.

Author action: We have performed validation test to improve our comparison section.

Reviewer#2,Concern#11: Your method could be applied to another extension of fuzzy sets such as ‘quasirung fuzzy sets‟. Include these in the future scope of the conclusion section by properly citing the following references. https://doi.org/10.1016/j.engappai.2022.105299;
https://doi.org/10.1007/s41066-021-00308-9

Author response: Thank you for suggestion.

Author action: We have mentioned this environment with these citations.

Reviewer#2,Concern#12: English writing should be improved considerably throughout the manuscript. 

Author response: Thank you for suggestion.

Author action: We have tried to improve it.

---

## [Decision Letter · Decision Letter 1]

12 Sep 2024

A Novel Approach to Decision Making in Rice Quality Management using Interval-Valued Pythagorean fuzzy Schweizer and Sklar Power Aggregation Operators

PONE-D-24-07498R1

Dear Dr.Binyamin,

We’re pleased to inform you that your manuscript has been judged scientifically suitable for publication and will be formally accepted for publication once it meets all outstanding technical requirements.

Kind regards,

PLOS ONE

Additional Editor Comments (optional):

I am happy to inform you that based on the reviewers comments, your paper now been accepted for publication.

Thank you

Reviewers' comments:

Reviewer's Responses to Questions

**Comments to the Author**

1. If the authors have adequately addressed your comments raised in a previous round of review and you feel that this manuscript is now acceptable for publication, you may indicate that here to bypass the “Comments to the Author” section, enter your conflict of interest statement in the “Confidential to Editor” section, and submit your "Accept" recommendation.

Reviewer #1: (No Response)

Reviewer #2: All comments have been addressed

2. Is the manuscript technically sound, and do the data support the conclusions?

Reviewer #1: Yes

Reviewer #2: (No Response)

3. Has the statistical analysis been performed appropriately and rigorously? 

Reviewer #1: N/A

Reviewer #2: (No Response)

4. Have the authors made all data underlying the findings in their manuscript fully available?

Reviewer #1: Yes

Reviewer #2: (No Response)

5. Is the manuscript presented in an intelligible fashion and written in standard English?

Reviewer #1: Yes

Reviewer #2: (No Response)

6. Review Comments to the Author

Reviewer #1: Authors revised the manuscript as per my suggestions. I have no any further suggestion for authors.

Reviewer #2: (No Response)

7. PLOS authors have the option to publish the peer review history of their article (what does this mean?). If published, this will include your full peer review and any attached files.

Reviewer #1: No

Reviewer #2: No

---

## [Editor Report · Acceptance letter]

10 Oct 2024

PONE-D-24-07498R1 

PLOS ONE

Dear Dr. Binyamin, 

I'm pleased to inform you that your manuscript has been deemed suitable for publication in PLOS ONE. Congratulations! Your manuscript is now being handed over to our production team.

Kind regards, 

on behalf of

Dr. Naeem Jan 

Academic Editor

PLOS ONE